# Insights into the Biology of *Leucocytozoon* Species (Haemosporida, Leucocytozoidae): Why Is There Slow Research Progress on Agents of Leucocytozoonosis?

**DOI:** 10.3390/microorganisms11051251

**Published:** 2023-05-09

**Authors:** Gediminas Valkiūnas, Tatjana A. Iezhova

**Affiliations:** Nature Research Centre, Akademijos 2, 08412 Vilnius, Lithuania; tatjana.jezova@gamtc.lt

**Keywords:** *Leucocytozoon*, haemosporidian parasites, life cycle, diversity, molecular characterization, birds, leucocytozoonosis, taxonomy, transmission, vectors

## Abstract

Blood parasites of the genus *Leucocytozoon* (Leucocytozoidae) only inhabit birds and represent a readily distinct evolutionary branch of the haemosporidians (Haemosporida, Apicomplexa). Some species cause pathology and even severe leucocytozoonosis in avian hosts, including poultry. The diversity of *Leucocytozoon* pathogens is remarkable, with over 1400 genetic lineages detected, most of which, however, have not been identified to the species level. At most, approximately 45 morphologically distinct species of *Leucocytozoon* have been described, but only a few have associated molecular data. This is unfortunate because basic information about named and morphologically recognized *Leucocytozoon* species is essential for a better understanding of phylogenetically closely related leucocytozoids that are known only by DNA sequence. Despite much research on haemosporidian parasites during the past 30 years, there has not been much progress in taxonomy, vectors, patterns of transmission, pathogenicity, and other aspects of the biology of these cosmopolitan bird pathogens. This study reviewed the available basic information on avian *Leucocytozoon* species, with particular attention to some obstacles that prevent progress to better understanding the biology of leucocytozoids. Major gaps in current *Leucocytozoon* species research are discussed, and possible approaches are suggested to resolve some issues that have limited practical parasitological studies of these pathogens.

## 1. Introduction

*Leucocytozoon* species (Leucocytozoidae, Haemosporida, Apicomplexa) are cosmopolitan blood parasites that only inhabit birds [1,2,3,4,5]. Species of this genus are unique among avian haemosporidians due to their inability to parasitize other vertebrates. This is not the case in species of haemosporidians belonging to the Haemoproteidae, Plasmodiidae, and Garniidae. Recent molecular phylogenetic studies have supported conclusions based mainly on phenotypic and ecological data regarding the close evolutionary relationships of leucocytozoids with species of other haemosporidian families, namely Haemoproteidae and Plasmodiidae [6,7,8,9,10]. This shows that *Leucocytozoon* species are important for better understanding the evolutionary biology of the Haemosporida, which includes the agents of malaria [1,5,7,11,12].

During the past 30 years, there has been remarkable progress in understanding the biology of apicomplexan parasites belonging to the Haemoproteidae and Plasmodiidae, including the sequencing of complete genomes of several species [13,14], many transcriptome assemblies [15], and rapidly developing molecular characterization (“barcoding”) at the species level [16,17]. However, there has been remarkably less progress regarding *Leucocytozoon* parasites [9,15,18]. Complete genomes of leucocytozoids remain unknown. The genetic diversity of *Leucocytozoon* species is significant, with over 1400 different lineages listed in the MalAvi database (http://130.235.244.92/Malavi/, accessed on 10 March 2023) [19]. However, only a small fraction of them (less than 2%) have been linked to the described parasite species, and some of these linkages are expected to be questionable or even incorrect [9]. Taxonomic work at the species level of these parasites remains negligible [20,21], as is the case with many other parasitic organisms [22]. This prevents the linkage of molecular data with available information about the biology of the parasites at the species level and ultimately hinders the application of new genetic knowledge in practical parasitological research.

The slow progress in understanding the biology of leucocytozoids is unfortunate. Species of *Leucocytozoon*, the agents of avian leucocytozoonosis, are not only cosmopolitan, but, contrary to haemoproteids (Haemoproteidae) and malaria parasites (Plasmodiidae), they are actively transmitted in countries with not only warm climates [3,4,5,10,23,24,25,26,27,28,29,30], but cold climates [7,31,32,33,34,35,36,37,38,39,40,41,42,43,44,45,46,47] and high mountains [48,49,50,51,52,53,54,55,56,57]. In other words, birds are under the threat of leucocytozoonosis nearly globally. These parasites cause diseases in poultry [1,3,27,58] and many species of wild birds belonging to most orders [4,5]. Thus, they are of broad importance in bird health. Only species of the orders Podicipediformes (grebes), Procellariiformes (albatrosses and petrels), Phoenicopteriformes (flamingos), Pterocliformes (sandgrouse), and Tinamiformes (tinamous) have not been found to be infected with leucocytozoids [3,4,59].

Notably, leucocytozoids are characterized by several unique biological features among haemosporidian parasites, such as the exceptionally rapid completion of their entire life cycle, transmission by black flies (Simuliidae), and unique host-parasite interactions on a cellular level [3,4]. Deeper research into the biology of leucocytozoids is important for better understanding the entire group of haemosporidian parasites, both from theoretical and practical perspectives.

While much knowledge exists and several reviews have been published on various aspects of *Leucocytozoon* species biology [1,2,3,4,60,61,62,63], they have not been updated in the past 15 years. This study aims to discuss available basic information and recent findings on the biology of avian *Leucocytozoon* species. Particular attention is given to currently recognized obstacles that hinder progress in better understanding these pathogens, and possible suggestions on how to overcome some of these issues.

## 2. Materials and Methods

This review mainly considers full-length papers published in peer-reviewed journals. Bibliographies of avian *Leucocytozoon* and other blood-inhabiting haematozoa [26,64,65,66,67,68,69,70,71] as well as reviews on haemosporidian parasites [1,4,5,23] were particularly helpful in collecting old publications (published before 1995). Recent articles were retrieved from online bibliographic databases PubMed, SCOPUS, and Google Scholar using the following keywords: ‘*Leucocytozoon*’, ‘birds’, ‘Akiba’, ‘Saurocytozoon’, ‘pathology’, ‘meront’, ‘schizont’, ‘Culicoides’, ‘black flies’, ‘Simuliidae’, ‘Simulium’, ‘sporogony’, and ‘exo-erythrocytic development’. The Boolean operators ‘OR’, ‘AND’, and ‘>’ were used to combine several terms. In all, approximately 1000 full-text articles and short reports were reviewed, but only those articles that explicitly provided data on the review subjects were cited. A total of 396 papers and publications were cited and incorporated into the references.

Parasite images were prepared from preparations deposited in museums and research institutions. Most of the preparations were from the collection of the Nature Research Centre (Vilnius, Lithuania), while some images were from preparations accessed in the collections of the Natural History Museum (London, UK), the International Reference Centre for Avian Haematozoa (Queensland Museum, Queensland, Australia), the US National Parasite Collection (National Museum of Natural History, Washington, DC, USA), and the Grupo de Estudio Relación Parásito Hospedero (GERPH) at the Department of Biology, Universidad Nacional de Colombia, Bogotá, Colombia. All accessed preparations or parasite images were studied using an Olympus BX61 light microscope (Olympus, Tokyo, Japan) equipped with an Olympus DP70 digital camera and imaging software AnalySIS FIVE (Olympus Soft Imaging Solution GmbH, Münster, Germany). Drawings were also provided to illustrate morphological features of the parasites for which high-quality photographs were not available. GenBank (National Center for Biotechnology Information, Bethesda, MD, USA) and MalAvi (Lund University, Lund, Sweden) databases were the main resources of information on genetic lineages and deoxyribonucleic acid (DNA) sequence information from these parasites.

## 3. Results and Discussion

### 3.1. Brief Outline of Leucocytozoon Life Cycle

Species of *Leucocytozoon* are haemosporidians characterized by the following basic traits: merogony or schizogony (asexual mitotic multiplication) takes place in cells of fixed tissues but never occurs in blood cells; malarial pigment (hemozoin) is absent at all stages of the life cycle; the sexual process and sporogony occur in Dipteran vectors; infection of vertebrate hosts occurs actively via bites of infected vectors.

The life cycles of *Leucocytozoon* parasites are obligately heteroxenous (Figure 1a–p), with parts that occur within the blood-sucking dipteran vectors and parts that occur only within birds [3,4,59]. Birds might be infected after exposure to the bites of infective vectors, which are black flies in all examined species, except for *Leucocytozoon* (*Akiba*) *caulleryi*. The latter parasite is transmitted by biting midges belonging to the Ceratopogonidae [4,23,72,73]. Available molecular phylogenetic data suggests that clarification is needed about the relationship of this species to the genus *Leucocytozoon* (see Section 3.3.1 for further discussion).

The following developmental stages and the sequence of their occurrence (Figure 1a–p) are found among *Leucocytozoon* species [1,2,3,4]. Infective stages are sporozoites (Figure 1f,g) and gametocytes (Figure 1j,k,o,p), which develop in vectors and birds, respectively (Figure 1f,g,j,k,o,p). In vectors, both the sexual processes (gametogenesis, fertilization) and the asexual multiplication (sporogony) occur (Figure 1a–e). Sporozoites develop in oocysts (Figure 1e) by means of mitotic (agamous) division. All vector stages (gametes, ookinetes, oocysts, mature sporozoites) occur extracellularly (Figure 1a–f).

Birds are intermediate hosts where only asexual multiplication (merogony or schizogony) occurs and merozoites develop (Figure 1i,n). The latter initiate the development of intracellular gametocytes (Figure 1j,k,o,p) that occur in blood cells and are infective for vectors. Unlike developmental stages in vectors, all parasite stages in birds are intracellular, but the final maturation of some tissue stages might be extracellular as occurs in megalomeronts of *L. (A.) caulleryi* in domestic chickens [73,74,75] and *Leucocytozoon* sp. (lineage lSTAL5) in owls [76]. It remains unclear how often the extracellular maturation of tissue stages occurs in leucocytozoids.

Sporozoites most likely initiate merogony in parenchymal cells of the liver (hepatocytes) where the first generation of meronts is most often found (Figure 1g) [31,77,78,79,80,81,82,83,84,85,86]. First generation meronts produce unicellular merozoites (Figure 1i) and the cytoplasmic fragments containing several nuclei—the so-called ‘syncytia’ (Figure 1l) [82,86,87,88]. Subsequent generations of meronts, which are initiated by merozoites from the hepatic meronts, might occur in various cells of fixed tissues (hepatocytes, renal epithelial cells as well as endothelial cells and macrophages in many organs). They were seen especially often in the spleen and lymph nodes.

The syncytia (Figure 1l) spread via the bloodstream to many organs, where they are phagocytized by macrophages or other reticular cells and produce megalomeronts (Figure 1m) [1,3,4,77,82,86,89,90,91]. Megalomeronts can be present within a wide range of host tissues and produce millions of merozoites.

Merozoites derived from meronts and megalomeronts are capable of invading various blood cells, including red blood cells, mononuclear white blood cells, and thrombocytes [81,82,85,91]. These parasites can also persist in avian hosts as dormant tissue stages, which may slow development during seasons that are unsuitable for vector-borne transmission, and produce merozoites leading to relapse-associated parasitemia when vectors are once again seasonally abundant [4,79,86].

Leucocytozoids have two types of exo-erythrocytic stages, namely the meronts (Figure 1h), which are relatively small (usually <80 µm in largest diameter), and the megalomeronts (Figure 1m), which are large (usually >80 µm and can exceed 300 µm in diameter) [31,77,78,79,81,82,83,84,85,86,87,88,89,92,93]. The latter are covered with a capsuler-like wall of host origin that may be more or less evident (Figure 1m) [2,4,86,94,95]. The host cell nucleus is usually visible in both types of meronts. It is markedly enlarged in megalomeronts (Figure 1m), but only slightly (if at all) in meronts [31,77,78,79,81,82,83,84,85,87,88,89,93]. The enlarged host cell nuclei of megalomeronts have been observed in many *Leucocytozoon* species and seem to be a characteristic feature of leucocytozoid development. However, host cell nuclei were not observed in some megalomeronts during recent histological studies [95]. This finding requires further research through examination of serial histological sections, which may be more likely to include the host cell nuclei.

*Leucocytozoon* spp. can persist in birds for many years or even for their lifespan [3,4,10,83,96,97,98], but the mechanism of persistence is not well understood. While persisting hypnozoite-like stages have not been documented in *Leucocytozoon* species or other avian haemosporidian parasites, such stages might occur in avian haemosporidians [99]. Persisting tissue stages (megalomeronts and meronts) have been found in *Leucocytozoon* species during seasons when vector borne transmission has ceased, indicating the involvement of exo-erythrocytic meronts in parasite survival during the seasons which are unsatisfactory for transmission (see Section 3.5 for further discussion).

Relapses occur in most *Leucocytozoon* species that have been studied. Based on current knowledge, they are exclusively due to the activation of persisting exo-erythrocytic stages and the production of merozoites that lead to an increase in parasitemia [82,89,96,97,98,100,101,102,103]. Unlike malaria parasites of the genus *Plasmodium*, recrudescences related to merogony in blood cells are absent in *Leucocytozoon* species. Rather, the increase in parasitemia likely indicates intensification of exo-erythrocytic merogony. This often occurs in synchrony with seasonal fluctuations in vector populations. However, circadian rhythms of parasitemia may lead to significant changes in the intensity of parasitemia during a 24-h cycle without a direct relationship to exo-erythrocytic merogony, possibly through the mobilization of sequestered parasites from blood vessels in the deep circulation [104,105]. Both relapses and circadian rhythms should be considered in research seeking a better understanding of the dynamics of parasitemia during leucocytozoonosis.

Uninuclear merozoites are responsible for the extensive spread of parasites within avian hosts (Figure 1i,n). Some merozoites from exo-erythrocytic meronts invade blood cells and develop into sexual stages (gametocytes or gamonts) (Figure 1j,k,o,p, Figure 2a–y and Figure 3a–w), while others initiate new cycles of exo-erythrocytic merogony (Figure 1h,i). Gametocytes are responsible for the production of gametes (Figure 1a,b) in vectors. Each macrogametocyte produces one macrogamete, and each microgametocyte normally produces eight microgametes [4,59]. Macro- and microgametocytes have recognizable sexually dimorphic characters under a light microscope. Macrogametocytes are characterized by intensely Giemsa-stained bluish cytoplasm and compact nuclei with distinct nucleoli and nuclear boundaries (Figure 2e–g,i,j,m–o,q,s,u,y and Figure 3b,c,e,g,i,k,m,o,q,s,u,v,w). By contrast, microgametocytes have a more pinkish cytoplasm, larger size, usually poorly defined boundaries of nuclei, and invisible nucleoli (Figure 2h,k,l,p,r,t,v,y and Figure 3d,f,h,j,l,n,p,r,t).

Differences in staining result from cellular structures that are significantly denser in macrogametocytes compared to microgametocytes [4,106]. Most described species produce gametocytes in either roundish (Figure 1j,k and Figure 2a–y) or fusiform host cells (Figure 1o,p and Figure 3a–w). If gametocytes from a single species develop in both types of host cells, those in the roundish host cells usually appear in early patency, while those in the fusiform host cells usually appear and predominate later [77,82]. Mature gametocytes in the circulation are essential for the infection of vectors, and gametocytes in both roundish and fusiform host cells are infective.

Mature gametocytes produce gametes (Figure 1a,b) in the midgut of vectors shortly after feeding on infected birds. Fertilization occurs extracellularly, resulting in zygotes (Figure 1c) that develop into worm-like motile ookinetes (Figure 1d). These ookinetes move towards the midgut epithelial layer, rounding up between epithelial cells or under the basal lamina and producing small oocysts (Figure 1e) [2,79,82,107,108,109]. The maturation of the oocyst (sporogony) terminates with the development of approximately 50–100 uninuclear elongate sporozoites (Figure 1f), which are released into the haemocoel. Some of these sporozoites reach the salivary glands, where they complete maturation. The sporozoite is an essential developmental stage for natural transmission during the vector blood meal.

Like all haemosporidian parasites, the prevalence of *Leucocytozoon* infection varies significantly among bird species, populations of the same species, and different localities [4,5,23,110]. Ecological factors such as season, climate, and topography ultimately govern the prevalence of infections, often resulting in wide differences within the same bird species from different localities [4,5,23,111,112,113,114,115]. Infection prevalence in juvenile resident birds before their first migration is a strong measure of local transmission.

Interestingly, the prevalence of *Leucocytozoon* infection in a bird population can be used in ornithology as a measure of philopatry in sites where black flies, the parasite-specific vectors, are absent, and transmission is completely interrupted. In other words, *Leucocytozoon* parasites can be used as biological tags in populational research in regions where vectors are absent, identifying bird-immigrants from other populations where transmission naturally occurs [116]. Examples of such sites include islands or semi-island ecosystems [4].

### 3.2. Remarks on Some Unique Haemosporidian Life Cycle Features

Among haemosporidian parasites, the following life cycle features are unique to *Leucocytozoon*, but their biological meaning remains insufficiently understood. First, viable sporozoites have been found in infected birds for several days, and even up to 11 days after a single injection [77,78]. This is the longest period known for persistence of haemosporidian sporozoites within vertebrate hosts. The role of such long sporozoite survival in birds remains unclear. It is possible that this adaptation allows the gradual penetration of sporozoites into host cells, leading to asynchronous merogony and a corresponding decline in the onset of pathology within host organs. It is worth noting that the long-term survival of sporozoites in avian hosts might lead to the amplification of *Leucocytozoon* DNA by PCR [117]. In other words, some solely PCR-based reports of *Leucocytozoon* infections and co-infections might be due to the amplification of DNA from persistent sporozoites, including those injected by vectors, that are unable to initiate development if they appear in the wrong (resistant or partly resistant) avian host. It is still unclear if many reported co-infections of *Leucocytozoon* lineages in naturally infected birds [7,51,118,119] represent competent or incompetent (abortive) infections at the sporozoite and/or exo-erythrocytic meront stage. It is worth mentioning that some abortive haemosporidian infections at the megalomeront stage might be virulent and even lethal when they appear in non-adapted avian hosts [4,94,120,121]. However, this health problem remains insufficiently studied among species of *Leucocytozoon*.

Second, *Leucocytozoon* gametocytes of some species induce the development of fusiform cytoplasmic processes in host cells (Figure 3a–w) [3,4,9]. These species include *L. danilewskyi*, *L. grallariae*, *L. lovati*, *L. neavei*, *L. neotropicalis*, *L. simondi*, *L. smithi*, and *L. sousadiasi* (Table 1). The host cell begins to elongate as the gametocyte grows (Figure 3a), rather than after the parasite has reached its full size. The host cell processes may be longer than the gametocytes themselves (Figure 3l,o,p,w). It has been suggested that such large cytoplasmic processes may be useful for sequestering the sexual stages in the deep circulation, which could be safer for gametocyte maturation if they partially escape from cellular immunity. The presence of gametocytes in the deep circulation may also benefit the parasite during periods when vectors (black flies) are inactive (at night) and the presence of fragile gametocytes in the peripheral circulation is unnecessary from the point of view of transmission.

Third, the development of young gametocytes markedly deforms infected host cells, and induces the enlargement of their nuclei and an increase in the amount of cytoplasm (Figure 2 and Figure 3). The same process also occurs in cells containing developing megalomeronts (Figure 1m). Such a close association indicates that the host cell is not simply a source of food for the parasite, as is the case with erythrocytes infected with malaria parasites, but rather that the parasite actively transforms the metabolism of the infected cells, turning them into a sort of “factory” producing material for parasite growth. This process is particularly pronounced during the development of megalomeronts, which cause a remarkable increase in the amount of chromatin and endoplasmic reticulum, as well as the number of mitochondria and other cellular structures in infected host cells [86,87,89,175]. Mitochondria have been observed within indentations of the nuclear envelope of gametocytes [176,177]. Although the metabolic processes and their association with host cell nuclei have not been studied in *Leucocytozoon* parasites on a molecular level, they are unique in haemosporidians and are worthy of further research.

Fourth, unlike species of *Plasmodium* and *Haemoproteus*, merozoites of *Leucocytozoon* parasites lack a pellicle, making gametocytes fragile and easily deformed in blood films [4,175]. Deformed gametocytes usually predominate in blood films, making the detection of non-deformed host-parasite complexes and species identification difficult. This lack of a pellicle may be related to peculiarities of metabolism in rapidly growing parasites, which can obtain nutrients, including those produced by the hypertrophied host cell nuclei, directly through the plasmalemma.

### 3.3. Diversity of Leucocytozoon Species

#### 3.3.1. The Classification Problems

Current species concepts have been thoroughly discussed in recent publications, including their role in organism dispersion, differentiation, and various host-parasite interactions [5,178]. This review is mainly focused on the taxonomic issues of parasite classification.

*Leucocytozoon* parasites are easily distinguishable from other haemosporidians by three primary features: (i) their ability to completely digest hemoglobin, resulting in the absence of pigment granules (hemozoin) in all blood stages; (ii) the absence of multiplication in the blood (only gametocytes develop in blood cells); and (iii) their marked influence on host cell nuclei. These features have been shown to have phylogenetic value, with the clade of most Leucocytozoidae species being distinct and well-supported in various phylogenetic analyses [7,8,11]. Another unique character among avian haemosporidians is the presence of sporogony in black flies [1,179]. Based on these characters, leucocytozoids transmitted by Simuliidae species have traditionally been classified in one genus, *Leucocytozoon* [1,59,61,180,181,182]. While the complete life cycles and details of the development in vectors and avian hosts remain unknown in most described species, the available data indicate that the major features of the phenotypic diversity of all studied parasites have similarities, suggesting provisional classification in one genus, *Leucocytozoon* [1,3,59].

Recent molecular studies have revealed that the genetic diversity of avian leucocytozoids may be even greater than that of the Plasmodiidae and Haemoproteidae [7,10,19,30,183]. Many molecular phylogenies have been proposed [7,8,9,10,11]. They are available in open access publications and are not repeated in this review. Several distinct clades have been identified within Leucocytozoidae, suggesting the existence of multiple subgenus-level parasite groups [9,10,93,184]. However, available information on parasites belonging to these groups is mostly limited to bioinformatic (DNA partial sequence) data, which precludes convincing taxonomic work at subgeneric levels of classification. More information about the biology and phenotypic characters of *Leucocytozoon* parasites belonging to different clades is needed to decide on the taxonomic rank and diagnostic features of different genetic lineages. The available data about tissue stages of *Leucocytozoon* spp. [76,86,93,95] show a diversity of morphological features and localization in organs which might be informative taxonomically, but remain unexplored in the classification of species of Leucocytozoidae because most species and genetic lineages have not been studied. Further research on the biology of leucocytozoids is necessary to better understand leucocytozoonosis, particularly the virulence and transmission patterns of pathogens belonging to different phylogenetic clades.

Phylogenetic studies suggest that *L. (A.) caulleryi* is more closely related to *Haemoproteus* (*Parahaemoproteus*) parasites than to *Leucocytozoon* species [6,7,8,9]. This is also supported by some phenotypic characters that unite *L. (A.) caulleryi* with *Parahaemoproteus*, but not with *Leucocytozoon*. For example, some species of *Parahaemoproteus* and *L. caulleryi* produce elongated merozoites in the first generation of their exo-erythrocytic meronts [185,186], whereas this is not the case among *Leucocytozoon* species [3]. The presence of a pellicle in the elongate merozoites of *L. caulleryi* is similar to the ultrastructure of merozoites of other *Haemoproteus* and *Plasmodium* species, but is different from the merozoites of *Leucocytozoon* species, which do not contain a pellicle [4,175]. In addition, both *H. mansoni* and *L. caulleryi* parasitize galliform birds and are transmitted by *Culicoides* spp. [73,187]. It is possible that the evolution of *L. caulleryi* is related to the *Culicoides*-transmitted species of avian haemoproteids due to these similarities and close phylogenetic relationships with *Haemoproteus* (*Parahaemoproteus*) spp. The investigation of mechanisms of hemoglobin digestion and hemozoin production in parasites in the subgenera *Parahaemoproteus* (hemozoin granules are visible) and *Akiba* (hemozoin is invisible)—a major difference between these organisms during development in vertebrates—might be helpful for answering this question.

Two species of *Leucocytozoon*-like parasites are known from reptiles. The genus *Saurocytozoon* was created for them [188]. *Saurocytozoon mabuyi* and *Saurocytozoon tupinambi* (Figure 2w,x) parasitize lizards [188,189,190,191,192]. Like species of *Leucocytozoon*, multiplication in blood cells was not seen in these protists, but blood stages (gametocytes) of *Saurocytozoon* species and their host cells (Figure 2w,x) are similar to *Leucocytozoon* species (Figure 2e–t) [189,192,193]. Malarial pigment (hemozoin) is absent in gametocytes of *Saurocytozoon* species. Due to these characters, *Saurocytozoon* was sometimes provisionally placed in the Leucocytozoidae, or even considered as a subgenus of *Leucocytozoon* [1,182,189,191]. However, limited experimental observations showed that *S. tupinambi* completed sporogony in laboratory reared *Culex pipiens* mosquitoes, developing large (up to 60 µm in diameter), *Plasmodium*-like oocysts with several germinal centers [191,193]. These characteristics differentiate *Saurocytozoon* parasites from both *Leucocytozoon* and *Akiba* species but unite them with species of the Plasmodiidae. However, the sporozoites were retained within oocysts and did not reach salivary glands, where they usually need to appear for their final maturation. It remains unclear if these findings represent normal development that would be found in a natural vector or are artifacts of development within an artificial experimental vector. Telford [193] classified *Saurocytozoon* with malarial parasites of the Plasmodiidae, however, the limited available information about these reptile parasites still seems insufficient for a conclusive taxonomic decision. DNA sequence information is absent for *Saurocytozoon* parasites, and their life cycles remain unknown in wildlife. Merogony in blood cells was not conclusively proved to be present in *S. mabuyi* and *S. tupinambi*, but it was seen in some infected lizards, possibly being a co-infection with other malaria parasites as is frequently seen in wildlife [193]. Further basic research—particularly on species diversity, molecular genetics and life cycles—is needed to clarify the taxonomic position and the origin of *Saurocytozoon* parasites and whether different species might be classified as branches of the Plasmodiidae, Garniidae, or even Leucocytozoidae.

#### 3.3.2. Leucocytozoon Species Taxonomy

Taxonomic research on species of avian leucocytozoids began in the late 19th century [132,194]. Nomenclatural difficulties related to the validity of the generic name *Leucocytozoon* and its type species have been the subject of long debates, which were finally resolved by a Decision of the Commission on Zoological Nomenclature [168]. The basis for this decision requires consideration of numerous nomenclature details (see [195,196,197]) and will not be repeated here. The authorship of the generic name *Leucocytozoon* belongs to N. Berestneff [194], and the authorship of its type species—the parasite of owls *Leucocytozoon danilewskyi*—was attributed to H. Ziemann [132], who described and illustrated this parasite. The designation of the main taxonomic characters of *Leucocytozoon* and the description of the first species were starting points for biodiversity research on avian leucocytozoids. Since then, many new species names have been suggested to distinguish *Leucocytozoon* parasites, mainly based on morphological features of their blood stages that were observed in different avian hosts. The first reviews of *Leucocytozoon* species were published by L. W. Sambon [64,198]. The early taxonomic studies were based mainly on a theoretical assumption that these organisms might be strictly specific to the bird species level [126,140,180,182,198,199,200,201,202,203]. As a result, nearly every report of the *Leucocytozoon* parasite in a new avian host was considered a new species, resulting in overcrowding of the literature with species names with insufficient descriptions and/or illustrations. This made it difficult or even impossible to distinguish named organisms from each other.

Further experimental observations have shown that some strictly host-specific *Leucocytozoon* parasites exist, but they likely are exceptions. Some examples are *L. caulleryi* in chickens and *L. smithi* in turkeys [4,204,205]. However, experiments have also shown that *Leucocytozoon* species that infect birds belonging to different genera of the same families also exist [206,207]. Moreover, some leucocytozoids successfully jump between birds of different families belonging to one order [108,208]. At present, there is no convincing experimental evidence to confirm that an isolate of the same species of *Leucocytozoon* can infect and produce gametocytes in birds belonging to different orders [31,61,83,205,208,209,210,211,212,213,214,215]. However, two interesting exceptions are worth mentioning. First, morphological and ecological observations suggest that the parasite of domestic chickens *L. schoutedeni* likely infects juvenile ostriches and causes severe disease, but adult ostriches are resistant [120,159,216]. This is an example of a so-called “child disease” when the parasite of Galliformes birds jumps to juveniles of Struthioniformes species. Nevertheless, there is still no molecular evidence to confirm this observation. In other words, it remains unclear if the parasites of domestic chickens and juvenile ostriches belong to the same lineage. Second, it is worth mentioning that a Common Loon (*Gavia immer*, Gaviiformes) was infected with the *Leucocytozoon* species in a rehabilitation center in the USA, likely from non-gaviiform birds [217]. These two observations demonstrate the possibility that *Leucocytozoon* species can complete their life cycle and produce gametocytes in birds belonging to different orders. However, this is unlikely to be a common phenomenon. The current taxonomy of *Leucocytozoon* species is based on relatively strong scientific information about the restriction of most parasite species and their lineages to birds of certain orders. This is supported by molecular data; cases when the same lineage of parasite was present in birds of different orders are rare [19].

Based on available experimental data, it is evident that there is no strict rule to distinguish *Leucocytozoon* species based on their host reports (species, genera, or families), supporting the notion that natural host range is not a valid taxonomic character [218]. In other words, reports of morphologically similar *Leucocytozoon* gametocytes and their host cells in different avian hosts can pertain to either the same or different parasite species. While the morphological characterization of blood stages provides some information about possible species identity, it cannot be used for definitive parasite identification due to the marked similarity of blood stages in many *Leucocytozoon* species. This is unfortunate because gametocytes are present in the peripheral circulation and are the most readily accessible stages of the life cycle for research. Other life cycle stages (exo-erythrocytic meronts, sporogonic stages) have been studied in a few *Leucocytozoon* species. Available information about their morphological features is fragmentary [76,86,93,95], and the taxonomic value of these features remains unexplored. The current classification of *Leucocytozoon* species by morphotypes (Table 1) can be considered an interim stage of taxonomic research. This scheme groups all non-distinguishable morphotypes present in birds of one order under the same name. Additional taxonomic research is needed to identify the true species diversity within these morphogroups.

The description of morphologically similar parasites as new species only because they are found in different families of birds—a common working taxonomic hypothesis during the past 30 years [163,164,165,166,167,219,220,221,222,223]—is no longer acceptable because it is not well-supported by available experimental observations (see reviews in [3,218]). Due to extensive ongoing reconstructions of bird classification at the level of both genera and families [224,225], the use of such an approach to guide *Leucocytozoon* taxonomy creates instability in the classification where changes in taxonomic position of avian hosts would inevitably require taxonomic reconsideration of the corresponding parasite species status. A more stable approach is to use host order—the most stable taxonomic level in bird classification—to guide the specificity of *Leucocytozoon* species.

In all, 45 morphotypes of *Leucocytozoon* have been distinguished (Table 1) and can be readily identified in the blood films of birds belonging to different orders. However, some of these morphotypes might be groups of closely related species or even non-related cryptic species. Both recent phylogenetic observations and the remarkable genetic diversity of avian leucocytozoids support this [7,19,51,93,153,169,183,184,226,227]. *Leucocytozoon* species taxonomy is still in its infancy and further information about the biology of these parasites is needed to clarify their taxonomic relationships.

#### 3.3.3. The Problem of Synonymous Species Names

Until proven otherwise, the names of all morphologically indistinguishable parasites in birds of the same order should be considered as synonyms of the same species, which have a priority in taxonomy (see reviews in [3,4]). This approach seems preferable given our current state of knowledge because it provides an opportunity to group *Leucocytozoon* parasites into distinct morphological groups. As experimental data often suggests, these are likely restricted in distribution to birds belonging to certain orders [31,61,83,205,208,209,210,211,212,213,214,215]. This is helpful for practical work during identification of the parasites using microscopic examination and visualization of their blood stages. Names of all morphologically indistinguishable parasites are synonymized with a name which has a priority in the nomenclature. Examples are the parasite of ducks and geese, *L. simondi* (synonyms are *L. anatis* and *L. anseris*); the parasite of passerines, *L. fringillinarum* (synonyms are *L. brimonti, L. bouffardi, L. cambournaci, L. chloropsidis, L. deswardti, L. dutoiti, L. enriquesi, L. gentili, L. icteris, L. molpastis, L. monardi, L. muscicapa, L. parulis, L. prionopis, L. pittae, L. roubaudi, L. thraupis, L. sturni, L. timallae, L. whitworthi*), and others [4]. It is important to note that recent molecular studies suggest that some of these morphotypes might be groups of cryptic species, sometimes not even closely related to each other [125,171,174,228]. In other words, the interim grouping of morphotypes by parasite species with nomenclatural priority within a bird order helps to clearly define potentially synonymous parasite descriptions. This can help to minimize the number of species names and guide future work that uses molecular phylogenetics to discover cryptic diversity.

A list of synonymous *Leucocytozoon* species names and those belonging to the categories of *species inquirenda* and *nomen nudum* has been published [4] and will not be repeated here. New possible synonyms, which were established after 2005, are listed in Table 1. It is important to note that due to the remarkable genetic diversity of *Leucocytozoon* parasites [7,19,33,51,93,183], many of the synonymous names may be validated in the future and can be used as a source for nomenclatural work. In other words, before establishing a new *Leucocytozoon* species name, the use of the available synonymous names should be prioritized.

#### 3.3.4. Morphological Characters of Blood Stages

Among *Leucocytozoon* parasites, parasite identification in blood films depends on the morphological characters of gametocytes and the peculiarities of their influence on host cells [1,2,3,4,180]. Contrary to other avian haemosporidians (species of Haemoproteidae, Leucocytozoidae and Garniidae), gametocytes of *Leucocytozoon* are remarkably similar in morphology (Figure 2 and Figure 3). In all species, the gametocytes appear as roundish or more or less oval bodies, which overlap in both form and size among many species [229]. If taken into consideration separately, these features are not valuable taxonomic characters for the development of keys for species identification. Among all leucocytozoids, gametocytes possess prominent nuclei, usually with readily visible nucleoli in macrogametocytes (Figure 2 and Figure 3). Pigment granules are absent in *Leucocytozoon* gametocytes and therefore, morphological features related to these granules—valuable taxonomic characters for *Haemoproteus* and *Plasmodium* parasites—cannot be used for taxonomic work. Volutin-like granules—non-refractive roundish cytoplasmic inclusions—are often present and prominent in the gametocytes of some species (Figure 2g,n and Figure 3f,g,m,q), but the taxonomic value of this character remains unclear and has been only poorly explored. However, contrary to other haemosporidians, developing gametocytes of *Leucocytozoon* induce marked and often taxonomically conservative changes in host cell morphology [Figure 2 and Figure 3] that can be used to distinguish species [4,180]. Among host cell characters, the following two features are most taxonomically informative.

First, host cell nuclei become markedly deformed and are often enlarged as gametocytes grow. The nuclei usually relocate to the periphery of the host cells and become closely appressed to gametocytes (Figure 2 and Figure 3). Importantly, the host cell nuclei often assume shapes, which are relatively conservative in certain parasite species or groups of morphospecies and important in species taxonomy [4,61,64,180,195]. For example, host cell nuclei associated with mature gametocytes assume various cap-like forms in species of the *L. fringillinarum* morphological group (Figure 2e–h), dumbbell-like forms in *L. dubreuili* (Figure 2i–l), and band-like forms in *L. majoris* (Figure 2m–p). The common morphological forms of host cell-nuclei in different *Leucocytozoon* species and their morphological groups are shown in Figure 2 and Figure 3.

Second, species of *Leucocytozoon* can be provisionally grouped in three morphological groups based on the shape of their host cells [4,61,180]. These are the species developing only in roundish host cells, which never produce spindle-like cytoplasmic processors (Figure 2); the species developing only in fusiform host cells, which have the spindle-like cytoplasmic processors (Figure 3s,t); and the species which develop both in roundish and fusiform host cells (Table 1). Many described species of *Leucocytozoon* are readily differentiated based on features of their host cells, but often not on morphological features of their gametocytes. In other words, *Leucocytozoon* species identification relies more on host cell morphology (the form of the host cell nucleus, the presence or absence of fusiform processors and their form) than on morphology of the parasites (gametocytes) themselves. This is a unique situation among haemosporidian parasites, indicating an important role of the host cell in growth and maturation of *Leucocytozoon* gametocytes and probably also in transmission.

The biological significance of the formation of the spindle-like processes on fusiform host cells remains unclear and the process of their development remains insufficiently studied [4]. Limited observations by electron microscopy suggest that compactly packed ‘bunches’ of parallel microtubules are involved in the formation of these spindle-like cytoplasmic processes. In some parasite species, these microtubules appear inside the nucleus of the gametocyte and eventually project into the parasite cytoplasm. They were described as being located in groups, which are arranged parallel to the long axis of the elongating parasite and the host cell [176,230]. The formation of spindle-like processes leads to the development of large host-parasite complexes which might exceed 40 µm in length [231,232]. Such host-parasite complexes might be important for the sequestration of the parasite in tiny capillaries of the deep circulation. Massive congestion of *Leucocytozoon* gametocytes has been documented in the deep circulation of various internal organs [93,233], possibly explaining rapid changes in gametocyte intensity (circadian rhythms) in the peripheral circulation that can occur over the course of a day [104,105]. Circadian rhythms might be an adaptation for the growth or persistence of gametocytes in the deep circulation. This might be beneficial for gametocyte survival, as is a case in *Plasmodium falciparum* and probably some other malaria parasites during gametocyte maturation in the extravascular spaces of bone marrow [12]. It is possible that the fusiform processors simplify the maintenance of gametocytes in the deep capillary circulation [93,230] due to the size of the host-parasite complex. However, precise mechanisms of gametocyte sequestration remain unknown.

Among all *Leucocytozoon* species, marked deformation and hypertrophy of host cell nuclei begins rapidly after the penetration of merozoites [4,82,85]. Early gametocytes usually appear adhered to the nuclei of host cells, and they often locate in an indentation of the nucleus (Figure 2c,d and Figure 3a) which assumes a species-specific morphology as the parasite matures. The close association of gametocytes and megalomeronts of *Leucocytozoon* with nuclei of host cells is remarkable (Figure 2 and Figure 3) and is a unique feature of this group of haemosporidian parasites (see also Section 3.1 and 3.2 for further discussion).

Recent molecular studies indicate that length and shape of cytoplasmic processes are important taxonomic characters for species identification and worthy of more attention in the classification of these parasites. For example, the parasites of diurnal raptors, *Leucocytozoon mathisi* and *L. buteonis,* belong to the *L. toddi* group and produce fusiform processes (Figure 3c–l). These species can be distinguished by the lengths and form of the processes [125], which are significantly shorter in *L. mathisi* (compare Figure 3c–f with Figure 3g–l). Different forms of fusiform processes in *Leucocytozoon* parasites are shown in Figure 3.

Recent molecular studies combined with microscopy revealed that co-infections of roundish and fusiform host cells from two different *Leucocytozoon* species might occur in the same host [9]. However, it remains unclear how often such co-infections occur in wildlife. The application of the commonly used general primers for the detection of partial sequences of the cyt*b* gene that are important for barcoding might not distinguish some co-infections [9,93,118]. This might lead to the incorrect conclusion that a single species can produce gametocytes both in roundish and fusiform host cells [9,153]. The application of several molecular methods in parallel (for example, the reading of the complete mitochondrial genome and the implementation of ordinary PCR-based testing using general primers) might identify the presence of a co-infection with two taxonomically distinct parasites [9] that would otherwise be confused as a single species [144]. Targeted molecular testing of such cases is needed for final taxonomic conclusions, but such studies remain rare [9]. This is a particularly important issue for understanding the pathology of *Leucocytozoon* species if co-infections are present [95,118,119,234].

Remnants of the host cytoplasm are usually seen around all growing gametocytes (Figure 2e,k,m,t) that develop in roundish host cells; therefore, this feature is not a useful taxonomic character [4]. Mature, fully grown gametocytes usually completely occupy infected cells and host cytoplasm is barely visible or invisible (Figure 2g,h,j,r,s,u,v).

It is worth noting that gametocytes of *Leucocytozoon* species are markedly fragile and often deformed during the preparation of blood films. Blood films should be thin, with blood cells not touching each other to help prevent deformation and secondary changes in the morphology of both parasites and host cells. The deformation of gametocytes is partly related to the absence of an inner interrupted membrane layer beneath the plasmalemma in merozoites and gametocytes with a subsequent loss of rigidity [4,175,235]. This should be taken into consideration when assessing the quality of blood films for taxonomic work if gametocytes of unusual form are observed.

### 3.4. Host Cells of Leucocytozoon Parasites

The origin of gametocyte host cells is difficult to determine based on morphological features because of their rapid and marked deformation by developing merozoites (Figure 2a–d and Figure 3a). Natural infections are typically low intensity and contain mainly mature gametocytes, making the microscopic observation of early stages of development difficult. The name *Leucocytozoon* was assigned to leucocytozoids due to the initial belief that the parasites develop in leucocytes [64,132,194,196,198]. This was shown to be true, but not for all parasite species. The rare microscopic observations of the early intracellular merozoites in blood cells and the application of histochemical methods showed that some species initiate development not only in mononuclear leucocytic cells but also in erythroblasts and erythrocytes [2,77,81,89,91,230,236,237]. Recent immunological observations show that thrombocytes are the main host cells in *L. macleani* (possible synonym is *L. sabrazesi*), the parasite of domestic chickens [144]. Recent microscopic and phylogenetic studies suggest that thrombocytes might be common host cells in some *Leucocytozoon* species. Furthermore, host cell origin might be predicted by phylogenetic analysis and can probably be used in taxonomy [20,238]. However, the origin of cells inhabited by gametocytes remains unknown for most *Leucocytozoon* parasites. The application of immunofluorescence diagnostic tools is needed for convincing conclusions, but is challenging due to the enormous genetic and antigenic variation in both wild birds and parasites, and the need to develop both host cell-type and host and parasite-specific antibody reagents [238].

Limited microscopic observations of naturally and experimentally infected birds show that primary meronts, which form from sporozoites, develop in hepatocytes, renal tubular cells, and endothelial cells of capillaries during *Leucocytozoon* infections [2,73,77,79,82,83,84,180]. The same is probably true for subsequent generations of meronts. Megalomeronts often develop in macrophages [2,82,90,91,180]. As is the case with gametocytes, the application of histochemical and immunological methods is needed for the better understanding of the origin of host cells during the development of tissue stages, but such studies have not been done for *Leucocytozoon*.

### 3.5. The Obstacles in Research on Exo-Erythrocytic Development

Exo-erythrocytic development has been investigated in a few *Leucocytozoon* species (see review in [86]) through the use of controlled experimental sporozoite-induced infections [31,72,77,78,79,82,84,89,98,102,175,186,239,240,241,242,243,244,245] and detailed histological examination of naturally infected birds with severe infections [76,81,83,85,90,91,93,94,95]. Different species of *Leucocytozoon* have different modes of development in tissues, but two distinct patterns seem to be common. A complete exoerythrocytic cycle might include: (i) the successive development of meronts (first) and megalomeronts (second) in the parasites of ducks (*L. simondi* [31,77,78,89,98,102,239,240], owls (*L. danilewskyi* [82]), and domestic chickens (*L. caulleryi* [72,73,75,186,241,242,246,247,248]); and (ii) only meronts, and not megalomeronts, develop in the parasites of turkeys (*L. smithi* [84,243,244,245]), penguins (*L. tawaki* [83]), and possibly some thrushes (*L. dubreuili* [79]) and New World blackbirds (*L. fringillinarum* [79]). Only fragmentary data about exo-erythrocytic development are available from most *Leucocytozoon* species, genetic lineages, and their avian hosts (see review in [86]), making it difficult to understand the general patterns of this process within this genus of parasites. The following findings are noteworthy.

First, the available experimental data shows that the exo-erythrocytic development of *Leucocytozoon* parasites is not strictly parasite species-dependent but can be different when the same species appears in different avian host species. For example, the same isolate of *L. simondi* develops both meronts and megalomeronts in ducks, but only meronts in geese after sporozoite-induced infections [206]. Gametocytes developed in both the ducks and geese in this study, indicating that they are competent hosts. Interestingly, the simplified exo-erythrocytic development in geese (lack of megalomeronts) was accompanied by lower parasitemia and virulence, as well as the absence of relapses [206,207]. These experimental studies are in accordance with limited observations of natural infections in other species which show that the megalomeronts of *L. sakharoffi* and *L. marchouxi* do not develop in all competent avian hosts [4,90,91,249].

Available information indicates that the formation of meronts (Figure 4a) is an obligatory tissue stage during *Leucocytozoon* development, but the appearance of megalomeronts (Figure 4b–j) might be skipped in some avian hosts. Megalomeronts might appear after the rupture of incompletely mature meronts, resulting in the development not only of unicellular mature merozoites (Figure 1i), but also of multinuclear cytoplasmic “islands” (syncytia) (Figure 1l), which can be phagocytized by macrophages and probably other reticular cells, in which they continue to grow and produce megalomeronts (Figure 1m). This might happen throughout the body of infected birds (Figure 4b–k), including in the liver, heart, brain, eyes, and sciatic nerves [3,4,61,77,180,246]. However, patterns of megalomeront emergence remain unclear in most *Leucocytozoon* species. The molecular mechanisms of this phenomenon are unknown.

A recent histological study combined with chromogenic in situ hybridization (CISH) described a new model of exo-erythrocytic development in *Leucocytozoon* parasites [76]. Only meronts of *Leucocytozoon* sp. (lineage lSTAL5) were seen in a naturally infected tawny owl *Strix aluco*. Large clusters of individual meronts at different stages of maturation and causing a tissue reaction were found developing in the lumen of blood vessels of kidneys and brain. The parasites seem to have initiated development in the endothelial cells of small capillaries, where they were visible as merozoites or early trophozoites. They then detached from their original location as the parasites grow, and flowed into the lumen of the blood vessel, where exo-erythrocytic meronts at different stages of maturation were observed developing asynchronously in big clusters. Due to the application of the genus-specific CISH diagnostics [76,94], there is no question about the generic identification of meronts, which certainly belong to *Leucocytozoon* sp. This finding is unexpected in regard to prior information about the exo-erythrocytic development of haemosporidians [2,3,86]. It shows that the exo-erythrocytic development of leucocytozoids is more diverse than is currently known. Further research on this subject may also lead to a better understanding of pathology during leucocytozoonosis. Recent studies suggest that exo-erythrocytic development might preferably occur in the kidneys of naturally infected accipitriform birds [93].

The application of CISH tools using parasite-specific probes opens new opportunities to link tissue stages found in wild birds with certain haemosporidian parasites [93,94,95,234,250,251,252]. This was difficult or impossible to do before the molecular era because co-infections of Apicomplexan parasites are common and even predominating in many avian hosts worldwide [4,7,9,10,23,38,40,42,58,105,110,118,253,254,255,256,257,258,259,260,261,262,263,264,265,266,267,268,269,270,271,272,273,274]. Recent molecular and microscopic studies show that the morphological diversity of haemosporidian exo-erythrocytic stages is unexpectedly diverse, particularly in *Haemoproteus* and *Leucocytozoon* parasites, which have historically been difficult or impossible to link to certain parasite species or lineages, even on genera levels, due to common co-infections [76,86,93,94,275,276,277]. Much new knowledge on this subject is expected as CISH is applied more widely. This technique can distinguish parasites of different genera in co-infected birds, thus opening new opportunities for understanding the exo-erythrocytic development and pathologies caused by haemosporidian parasites.

Further experimental observations, combined with molecular and experimental research are required for determining patterns of the exo-erythrocytic development in leucocytozoids. The parasites of domestic birds are convenient targets for such studies due to their practical significance, their susceptibility to haemosporidians, the relatively simple logistics of experimental design and permitting, and the relative ease of obtaining parasite and host material (including materials and templates for immunological tests).

### 3.6. Challenges of the Molecular Characterization of Leucocytozoon Parasites

Approximately 45 *Leucocytozoon* morphospecies can be distinguished, however, only 13 (or 29%) have been characterized by molecular methods (Table 1). Of over 1400 detected *Leucocytozoon* lineages [19], most are unidentified to the level of species or their species identity is questionable. This is unfortunate because the application of molecular markers can speed research on various aspects of haemosporidian biology, and is particularly valuable for the identification of various stages of the life cycle in avian hosts and vectors [18,21,76,86,93,105,184,277,278,279,280,281]. Details about the use of molecular methods in studies of avian haemosporidian parasites were recently reviewed [282], including the description of the MalAvi database, the use of barcoding, species limits, the selection of primers, and the peculiarities of molecular work. This information is not repeated in this review. A number of obstacles need to be overcome to make full use of molecular tools for the characterization of *Leucocytozoon* species.

First, due to high genetic diversity [7,9,10,19,41,169,183,283,284], the number of *Leucocytozoon* species is expected to be larger than the number of currently described parasites. This is true for all genera of haemosporidians [5,16,17,19,86,285]. However, the *Leucocytozoon* parasites are exceptional in this regard due to particularly large genetic distances among partial cyt*b* gene sequences for even closely related species. Genetic divergence of over 10% between parasites within the same phylogenetic clades seems to be a common and characteristic feature of *Leucocytozoon* [9,10,29,93,184]. This indicates that the number of evolutionary independent entities might be enormous among leucocytozoids. Additionally, their molecular detection might be exceptionally challenging because of difficulties in designing general primers for DNA amplification. Such primers would be particularly useful for detecting undescribed parasites and expanding biodiversity research. For example, some lineages of the South American leucocytozoids are impossible to detect using the primers which were designed using DNA sequence information from *Leucocytozoon* species that parasitize passeriform birds in Europe [9]. Numerous morphologically similar *Leucocytozoon* parasites have been described that require better molecular characterization in relation to their distribution by species of avian hosts. Methodological problems of haemosporidian molecular diagnostics in wildlife have been discussed elsewhere [9,282]. The following taxonomic challenges are less well known but worthy of attention.

The rules of international nomenclature give priority to the first described species in situations where multiple synonymous descriptions and names need consolidation [197,286]. Ideally, the molecular characterization of leucocytozoid species should focus first on samples from the vertebrate type host, preferably collected from the type localities or locations which are close to the type localities. This is especially important because of the enormous genetic diversity of leucocytozoids, even in closely related avian hosts [19]. Such molecular characterizations should be a starting point for the further development of *Leucocytozoon* species taxonomy using DNA sequence information. While substantial progress has been made with haemosporidian parasites belonging to the Haemoproteidae and Plasmodiidae [16,17,19], significant gaps are present for leucocytozoids.

Unfortunately, the DNA of most *Leucocytozoon* species which have priority in nomenclature have not been sequenced, and their genetic lineages remain unknown (Table 1). This is a significant taxonomic obstacle for identifying species by molecular barcodes and advancing biodiversity research, including the recognition and description of new *Leucocytozoon* parasites. For example, there is no DNA sequence information about the parasite of owls *L. danilewskyi* from its type host (date of the description is 1898; the type host is *Athene noctuae*; the type locality is Italy) or the prevalent parasites of passerines *L. majoris* (1902, *Parus major*, France), *L. berestneffi* (1908, *Pica pica*, southern Russia), *L. fringillinarum* (1910, *Fringilla coelebs*, UK), *L. dubreuili* (1911, *Turdus* sp., Vietnam), the parasite of diurnal raptors *L. toddi* (1908, *Kaupifalco monogrammicus*, Congo), and most other parasite species described in the 20th century (Table 1). The absence of information about the genetic lineages of these species, which have priority in the nomenclature and taxonomy, limits further taxonomic advances using molecular markers. For example, the genetic lineage with GenBank accession FJ168564 was assigned to *L. fringillinarum* based on the morphology of visible gametocytes and their host cells. However, this lineage was detected in *Pipilo chlorurus* (Passerellidae) sampled in America, while the original description of *L. fringillinarum* came from *F. coelebs* (Fringillidae) sampled in Europe. The parasite from *P. chlorurus* might be *L. fringillinarum*, but also might be a genetically different cryptic species with similar gametocytes and host cells. In other words, gametocytes from *P. chlorurus* are similar morphologically to *L. fringillinarum*, but there is no convincing evidence that sequence data deposited in GenBank belongs to this parasite. The genetic lineage remains unknown in the type host, the chaffinch *F. coelebs* [238]. The lineage found in *P. chlorurus* might be a different parasite and is currently considered to be a morphological synonym of *L. fringillinarum*. Because material originated from non-type hosts far away from the type localities, the same shortcomings in the molecular characterization of *Leucocytozoon* species are expected in GenBank accessions FJ168563, MG734971, MG734973, and many others (see Table 1 for further comments).

Second, the molecular characterization of *Leucocytozoon* species is often confused because of common co-infections of parasites belonging to the same genus [7,10,38,40,42,105,118,119,254,256,257,259,260,261,262,263,268,270,272,273,274]. Microscopic examination of blood films from many naturally infected birds shows that infection with a single species of *Leucocytozoon* is rare. This is particularly true in passerines [9,253]. For example, chaffinch is often co-infected with parasites morphologically similar to *L. fringillinarum*, *L. majoris*, and sometimes *L. dubreuili*—none of which have been barcoded from their type hosts. It is challenging to link a detected lineage to a specific parasite species in co-infected birds [282], and it would be valuable to develop methods to detect single *Leucocytozoon* infections and link specific lineages to certain parasite species. There has been no significant progress in the molecular characterization of leucocytozoids, mainly because accurate linkage of observed parasites with their genetic lineages requires the extensive targeting analysis of many samples both by microscopy and PCR-based testing using different protocols. Recent work indicates that a combination of commonly used PCR-based protocols and cloning of the mtDNA genomes, as well as second generation sequencing, might be helpful for distinguishing co-infections and identifying unique parasite lineages [7,9,282].

Third, it was established experimentally that some *Leucocytozoon* species produce gametocytes in both roundish and fusiform host cells (Table 1). These host-parasite complexes are easy to distinguish in blood films (Figure 2 and Figure 3). For example, experimental observations showed that both types of host cells develop and belong to the same species in *Leucocytozoon simondi* and *L. danilewskyi* infections [77,82]. In other words, the gametocytes developing in roundish and fusiform host cells should be the same genetic lineages in these species, but this has not been proven by molecular techniques. By contrast, gametocytes in roundish and fusiform host cells might be from different species in some *Leucocytozoon* infections, as is the case during natural infections of *L. grallariae* and *L. neotropicalis* [9]. In other words, it is not always possible to assign gametocytes in roundish and fusiform host cells to one parasite species even if they present in one sample [144]. This is particularly true during natural infections. Furthermore, it might not always be possible to solve this question using one or even several PCR-based protocols due the insufficient sensitivity of available general primers and the high probability of the presence of cryptic species which seem to predominate in *Leucocytozoon* [7,9,93]. The remarkable genetic diversity of *Leucocytozoon* lineages in species of the Anatidae, in which only one distinct morphospecies has been described, suggests the existence of several species with gametocytes similar to *L. simondi* [41,103]. The same is likely true in many other *Leucocytozoon* morphospecies, for example *L. toddi* [93,226], as well as the morphological groups containing *L. fringillinarum, L. majoris*, and *L. dubreuili* lineages [184,228].

Fourth, the intensity of parasitaemia is usually low (usually <1 gametocyte per 10,000 red blood cells) during *Leucocytozoon* infections. This is an obstacle for the identification of *Leucocytozoon* morphospecies when combined with the marked fragility and deformation of the parasite-host complexes in blood films that may affect morphology. Because multiplication in the circulating blood cells is absent in *Leucocytozoon* parasites—a key difference from *Plasmodium* species—birds usually cannot be infected by subinoculation of infected blood, and experimental transmission must rely on use of a vector. A rare exception is the case when the invasive merozoites from tissue stages are present in the circulation [4]. This limits experimental efforts to obtain a full range of blood stages for taxonomic purposes as is commonly done in studies of avian malaria [99]. The preparation of several blood films for the collection of a representative number of blood stages for parasite morphological identification is recommended when hosts are sampled for studies of *Leucocytozoon*.

It is clear that the morphology of blood stages cannot provide consistent answers about the diversity of *Leucocytozoon* species because of their remarkable cryptic speciation. A combination of phenotypic and molecular approaches in species delimitation, including the investigation of tissue and vector stages, is essential for reliable comparative studies. However, investigations of life cycles of parasites are time consuming. Clearly, DNA sequence information will accumulate significantly faster than it can be used to describe species by combining morphological and molecular approaches. An interim solution that has proven to be a productive approach for studies of avian *Plasmodium* and *Haemoproteus* parasites is to assign these diverse genetic lineages distinctive acronyms [16,17,19,76,287,288].

### 3.7. The Obstacles in Leucocytozoon Parasite Vector Research

The most thoroughly described and investigated *Leucocytozoon* species are transmitted by black flies of the family Simuliidae [3,4,179,213,289]. Available experimental data show that the same species of black fly can transmit different parasite species, and the same parasite species can be transmitted by many species of black flies belonging to different genera and subgenera [3,80,81,82,83,107,108,215,290,291,292,293,294,295]. It seems that restrictions for the use of certain species of black flies for transmission are ecological rather than physiological in *Leucocytozoon* species [2,290,296,297,298]. For example, the black flies preferably feeding in tree canopies are susceptible to *L. simondi*—the parasite of anseriform birds—but they are ecologically isolated in part from birds that spend much of their time at ground level. This should be taken into consideration in epidemiological studies aimed at the determination of natural vector species.

Methodologies for experimental infection of free-leaving black flies with *Leucocytozoon* parasites were developed and described using relatively large sentinel birds (ducks, turkeys, and others) [3,213,292,296,299], but are much less effective when working with small birds that are less attractive to vectors. Significant effort is necessary for the collection of even a few engorged insects using small-size sentinel birds, for example, the small passerines [300]. With species of *Plasmodium* and *Haemoproteus*, relatively easy and cheap methods for the colonization of some vectors, including mosquitoes (Culicidae), biting midges (Ceratopogonidae), and louse flies (Hippoboscidae) led to significant advances in experimental vector research [281,301,302,303]. It is also relatively easy to infect wild-caught biting midges and mosquitoes experimentally by direct feeding on infected birds [279,304]. Black flies are significantly more difficult to colonize because the larvae of most species require oxygen-rich, flowing water for development [305,306]. Such conditions are challenging to design and maintain under laboratory conditions [307,308,309,310], and colonized black flies have never been successfully used in *Leucocytozoon* vector research. A possible approach for experimental research with black flies is to collect their mature pupae in the wild, use them as a source of adult flies in the laboratory, and then infect these insects by feeding on birds with patent parasitemias. Survival of the flies in captivity is usually low but can be increased by maintaining them in cool conditions (about 14–16 °C) and complete darkness, which prevents active movement of the insects, and increases their survival time. *Leucocytozoon* parasites complete sporogony at similar temperatures within 4–6 days [4,109]. Such experiments also might be useful for obtaining material for morphological and molecular studies of the development and host-parasite interactions of *Leucocytozoon* within its black fly vectors. It is worth noting that many species of ornithophilic insects prefer to breed in small, clean forest streams [305,306], which may be good sources for collection of pupae when designing laboratory experiments with avian leucocytozoids.

The determination of natural vectors of *Leucocytozoon* might be approached using a methodology that was successfully developed for studies of the vectors of *Haemoproteus* [311]. Black fly females can be collected in the wild using traps [312,313] or simple nets. Small salivary gland smears (about 0.5 cm^2^) can be prepared from the thorax of dissected flies on glass microscope slides [314]. Remnants of the dissected thoraxes are then fixed and tested by PCR-based methods for detecting the presence and identity of *Leucocytozoon* lineages. Salivary gland smears of the PCR-positive samples can then be stained with Giemsa and examined by microscopy for the possible presence of the sporozoites. The total number of slides to examine should be few because of the predominantly low natural prevalence of *Leucocytozoon* infections in wild simuliids. This method was effective for determining vectors of *Haemoproteus*, as well as for studying the natural feeding preferences of biting midges and black flies [314,315,316,317].

Some field observations and recent molecular data indirectly suggest that biting midges of the genus *Culicoides* might be involved in the transmission of leucocytozoids of accipiriform birds [10,112], however, there is no convincing evidence for this hypothesis. Vectors of *Leucocytozoon* species parasitizing accipitriform birds have never been investigated experimentally. Due to the involvement of *Culicoides* species in the transmission of *L. (A.) caulleryi* [4,73], this hypothesis is worth attention.

### 3.8. Puzzles of the Geographical Distribution

*Leucocytozoon* species are cosmopolitan and actively transmitted in both cold and hot climates. High prevalence of these parasites (for example, close to 100% in some species of ducks and birds of prey) has been reported globally, including in environments located far above the Arctic Circle [4,24,31,32,39,44,45,46,118,261,318,319,320]. It is worth noting that leucocytozoids are the dominant blood parasites in some areas of the Nearctic region [24,223,321,322]. These parasites are prevalent in New Zealand [35,92,160] and in cold mountain sites located at altitudes greater than 3000 m above sea level [48,49,50,119,228]. Leucocytozoids are also prevalent in the hot tropical regions of the New World [5,25,69], where they flourish in some bird species with prevalences of over 50% in passerines of some families [1,4] and prevalences of close to 20% in domestic chickens [29]. This broad geographic and ecological distribution illustrates the global availability of vectors (species of the Simuliidae) and the vectorial capacity of many black fly species belonging to different subgenera. It also illustrates the wide range of temperatures that can support the relatively rapid development of salivary gland sporozoites. For example, sporogony of *L. simondi,* a parasite of ducks and geese, is completed within 4–5 days at 15 °C [4]. This tolerance to low temperatures helps to explain the broad distribution of leucocytozoids in cold environments, including mountains. It is interesting to note that the sporogony of leucocytozoids is extremely rapid at high temperatures. Sporozoites of *L. caulleryi* can appear in the salivary glands of vectors in as little as 2–3 days after a blood meal at temperatures of 25–30 °C [323]. The development of leucocytozoids within birds is also extremely rapid in some parasite species, with the completion of exo-erythrocytic development and the appearance of mature gametocytes in the peripheral circulation within 4–5 days after the inoculation of sporozoites [87]. The entire life cycle of *L. simondi*—from infection of vectors to the development of infective gametocytes in birds can be completed within a week. This is the fastest known time for the completion of a life cycle among haemosporidians [1,4]. The rapid completion of the life cycle and the ability to complete sporogony in many species of the Simuliidae [3,290] have likely contributed to the global spread of leucocytozoonosis.

Based on available information about life cycles, the same species of *Leucocytozoon* might theoretically be expected to have a global area of transmission in cosmopolitan avian host species. However, this does not occur, indicating the existence of still insufficiently understood mechanisms for the restriction of parasite distribution. The following examples of the remarkably patchy geographical distribution of *Leucocytozoon* species remain unexplained in some cosmopolitan avian hosts.

First, ducks and geese (Anseriformes), including a few domesticated species, are globally distributed, but are infected with *L. simondi* predominantly in the Holarctic north to 42° N [58]. This parasite is prevalent in anseriform birds in the northern Holarctic [3,4,24,31]. The few records that have been reported outside of this region may likely be in overwintering migratory birds from northern populations [3,4,25,28,34]. It is worth noting that low rather than high temperatures seem to be optimal for sporogony of some strains of *L. simondi* [31]. While this adaptation contributes to the transmission of this pathogen in cold ecosystems, it is unlikely to be the main factor restricting the spread of *L. simondi* in the tropical ecosystems. The absence of active transmission of *L. simondi* in the tropics remains unexplained.

Second, *L. smithi*, the parasite of wild turkeys *Meleagris gallopavo*, is actively transmitted only in North America. While domesticated turkeys are present worldwide, there are only a few reports of the introduction of *L. smithi* in France (the site of this parasite discovery and original description), South Africa and Ukraine [27,157,324,325]. Natural transmission of the parasite has not become established in these areas for reasons that are not understood. It seems that the natural nidi of *L. smithi* infection are essential to maintain the transmission, and such nidi are present only in North America where infected wild turkey populations exist.

Third, domesticated ostriches *Struthio camelus* breed globally, but *L. struthionis* is reported mainly in South Africa, with a few reports in tropical Africa [4,120,159,216]. It seems that transmission of this parasite is restricted to sub-Saharan ecosystems.

Fourth, *L. (A.) caulleryi*—the parasite of the cosmopolitan domestic chicken *Gallus gallus*—is actively transmitted, mainly in South and Southeast Asia [1,72,73], with one report known in south Kazakhstan [326], and recent reports in Egypt [327,328]. It is reasonable to expect the global spread of *L. (A.) caulleryi* because: (i) the avian host is cosmopolitan and vectors (biting midges of the Ceratopogonidae) are globally distributed [4,5,73,329]; (ii) the same parasite strain can complete sporogony in many species of these flies [3,73]; and (iii) sporogony of leucocytozoids is possible under a wide range of temperatures [72,73,323]. However, this has not occurred, and leucocytozoonosis caused by *L. caulleryi* remains remarkably patchy in global distribution. It seems that domestic chickens obtained this infection from the wild red junglefowl where their ranges overlap in South Asia [4]. Reports in other galliform birds are absent. The parasite is virulent in domestic chickens, and the infection is often lethal in non-adapted chicken breeds [1,72,327,330,331,332]. Experimental studies have shown that a high degree of immunity develops in domestic chickens that survive acute infections. Relapses are absent in these birds, resulting in the absence of viable gametocytes in the circulation that can infect potential vectors [73,333,334,335]. The absence of circulating gametocytes may help prevent the introduction of *L. caulleryi* when chickens are moved to other regions of the Earth. In situations where the persistence of gametocytes in the peripheral circulation is either brief or absent, infections may persist within the vector population instead. For *L. (A.) caulleryi*, sporozoites remain viable for approximately a month after maturation within their vectors [4,323]. This might be enough time to span short periods of time that are not suitable for transmission, but this has not been sufficiently investigated. It remains unclear how often persistence of these parasites in vectors occurs in other haemosporidians, but it may be an important evolutionary adaptation, particularly during the winter where hibernating dipterans may persist until the spring. Opportunities for the human-mediated transportation of infected vectors have increased as a result of the globalization of trade [336,337,338]. This might explain some recent reports of *L. caulleryi* outside of South Asia [327,328], and highlights ongoing efforts by many countries to improve biosecurity (e.g., New Zealand).

Limited molecular ecology data on the distribution of *Leucocytozoon* lineages in wild passeriform birds suggest that novel introductions of parasites into resident birds during seasonal migrations are rather rare events [34,278,339]. However, as seen with the case of leucocytozoids in domesticated birds, the mechanisms preventing the spread of these pathogens globally remain insufficiently understood. While narrow host specificity may help reduce the spread of these parasites by migrants [5], it does not explain the remarkably patchy geographical distribution of certain *Leucocytozoon* lineages in some globally distributed bird species. These data show that more detailed examinations of host-parasite interactions in certain host-parasite associations are needed to unravel geographical anomalies in the distribution of these parasites. This information may be important for the development of strategies for vector-borne disease prevention. For example, it is difficult to explain why *Plasmodium juxtanucleare*—a malarial parasite of the domestic chicken—is globally distributed in the tropics, but *L. caulleryi* is not. Both the domestic chicken and the vectors of *P. juxtanucleare* and *L. caulleryi* (*Culex* mosquitoes and *Culicoides* biting midges, respectively) are present globally, but global geographical distribution of the parasites is different. Perhaps, the long-lasting relapsing infections that are characteristic of *P. juxtanucleare* enhance transmission, while transmission of *L. caulleryi* is prevented by the rapid elimination of gametocytes from the blood and the absence of relapses in recovered chickens [72,73,333].

Exceptionally low prevalence and diversity of *Leucocytozoon* parasites in the lowlands of South America remains a long-lasting puzzle [3,4,340]. The Amazon basin is a good example [341,342]. Both microscopy and molecular diagnostics have revealed that leucocytozoids are rare in this region. Numerous lineages of the parasites appear regularly in Amazonia with migratory birds from the Nearctic, but transmission is either absent or probably only recently established and remains neglectable [339,341,343,344]. Simuliid vectors are abundant in South America [52,345] and *Leucocytozoon* spp. transmission is active in mountainous regions [171,346], indicating that susceptible vectors and avian hosts are present in South America. However, transmission appears to be interrupted in the lowlands. The susceptibility of South American simuliid flies to *Leucocytozoon* parasites and the peculiarities of sporogonic development have not been studied in the Amazon. Similarly, patterns of bird-vector associations are also insufficiently known. It is difficult to rule out the possibility that lineages of *Leucocytozoon* of northern origin may infect the local birds, but cannot complete exo-erythrocytic development in resident avian hosts that may include endemic species. This might lead to severe leucocytozoonosis and mortality from the tissue stages of abortive infections, as has been described during leucocytozoonosis in domestic ostriches, pigeons, and parrots [120,347,348], as well as with avian *Haemoproteus* infections in non-adapted avian hosts [121]. Detailed experimental and field studies of avian hosts and their susceptibility to parasite lineages are needed for a better understanding of the biology of avian leucocytozoids and the factors that determine their geographical distribution.

### 3.9. Pathogenicity and Related Health and the Host-Parasite Association Issues

*Leucocytozoon* parasites are pathogenic both in blood-sucking dipteral insects and avian hosts [3,4,217]. However, virulence varies remarkably among different parasite species, their isolates, and their avian host species.

The life span of blood-sucking black flies (Simuliidae) in the laboratory is inversely proportional to the intensity of *Leucocytozoon* parasitemia in birds on which they feed [83,290,349]. Most black fly females die within 24 h after taking blood meals from birds with high parasitemia infections with *L. simondi* and *L. tawaki*. Similar results have been obtained in experiments with mosquitoes and biting midges during experimental *Haemoproteus* infections [279,350]. Mortality was associated with damage caused by massive migration of ookinetes from the blood meal throughout the body of the exposed insects.

Low chronic parasitemia often appears benign in wild birds that are observed in captivity, but the consequences of severe primary sporozoite-induced infections remain insufficiently understood in wildlife [3,4,217]. Some long-term field observations and case reports suggest that leucocytozoids might be important health problems [4,351,352]. Unfortunately, the pathogenicity of only a few parasite species has been studied in detail. These include parasites of anseriform and galliform birds, pigeons, and ostriches (see reviews in [3,4,353]). Because of the current restrictions on the use of wild birds for research, experimental observations remain scarce. The most convincing data on the pathogenicity of leucocytozoids comes primarily from research published over 30 years ago, with only a few recent case reports and field studies.

Some *Leucocytozoon* species, especially *L. simondi* in ducks, *L. smithi* in turkeys, *L marchouxi* in doves and pigeons, *L. caulleryi* in domestic chickens, *L. struthionis* in ostriches, and *Leucocytozoon* sp. In falcons have been demonstrated to be virulent and cause severe leucocytozoonosis and mortality [72,73,77,155,156,208,213,231,242,244,331,347,354,355,356,357,358,359,360,361,362,363,364,365,366]. The most recent cases of mortality of domestic ducks due to leucocytozoonosis are from Sweden [367]. Clinical signs of leucocytozoonosis are usually nonspecific and may not be apparent [353]. Anaemia, lethargy, loss of appetite, breathing difficulties, debilitation, depression, erratic flight, lack of coordination, diarrhea and defecation with greenish feces are frequently observed during severe leucocytozoonosis caused by multiple parasite species. Convulsions and paralysis may occur before death during acute infections. The death rate in juveniles less than one month old is often reported to be high, while it usually decreases in birds of greater ages. Adult birds tolerate the disease more easily and usually survive. Necropsy usually reveals emaciation, dehydration, congestion in the lungs, liver, and spleen, and enlargement of the spleen and liver in cases of severe leucocytozoonosis [245,353,368]. Some species, for example, *L. macleani* (possible synonym is *L. sabrazesi*) and *L. schoutedeni*, cause relatively mild diseases in chickens, but a decrease in the productivity has been reported [3,4,369].

Pathological changes are often associated with the development of megalomeronts, which sometimes reach up to 500 µm in diameter [72,88,90,91,347,358,370,371,372]. An inflammatory reaction has been observed around ruptured megalomeronts and haemorrhages might develop. Neurological complications including cerebral paralysis have been reported if the parasites develop in the brain.

The most severe pathological consequence of leucocytozoonosis is anaemia [1,72,373,374]. This develops from the direct destruction and removal of infected erythrocytes and the appearance in the blood plasma of a so-called anti-erythrocyte factor that is poorly understood [3,373]. There is a pronounced increase in the osmotic fragility of uninfected erythrocytes and erythrocyte haemolysis during leucocytozoonosis, but consequences of the resulting anaemia remain unclear in wildlife. It is important to note that severe pathology may be due to gametocytes, which initiate development of large host–parasite complexes (40 µm and even longer). For example, *L. smithi* gametocytes block the alveoli and congest the lungs during high parasitemia, which leads to distortions of respiratory functions and the development of pneumonia-like symptoms [3,4,375]. Recent molecular studies using chromogenic in situ hybridization with genus-specific probes have provided additional support of these observations by simplifying the visualization of gametocytes in histological sections of organs [93].

While similar pathology has been described in wild birds [55,163,195,233,348,365,376,377,378], research on the effects of leucocytozoonosis in wild populations remains fragmentary. It has been proved experimentally that sporozoite-induced infection of *L. simondi* leads to the death of immature wild anseriform birds (*Anas rubripes, A. platyrhynchos,*) [379]. This parasite is also virulent in swans *Cygnus olor* [380]. While published case observations on the pathogenicity of leucocytozoids in wild birds are rare, they are dramatic. For example, neurological symptoms and anaemia were readily apparent in one emaciated juvenile *Anas platyrhynchos* infected with *L. simondi* that was recovered from the wild in Canada and necropsied [381]. Up to 400 megalomeronts were found in the histological sections of the brain within an area of 1 cm^2^. Blockage of blood vessels and hemorrhages were numerous around ruptured megalomeronts. Epizootics caused by *L. simondi* have been reported in ducks [289,355] and immature geese *Branta canadensis* [382] in the USA. An outbreak caused by *Leucocytozoon* sp. in a colony of weaver finches *Ploceus jacksoni* was described in Kenya [1]. Infection by *L. danilewskyi* (synonym *L. ziemanni*) caused a decrease in clutch size among infected *Aegolius funereus* in years when access to food resources was limited [383]. Increases in the intensity of *Leucocytozoon* spp. infection were reported in males of *Parus major* that spent more energy during the period of reproduction [384]. It is important to note that long-term studies of populations of Eurasian Sparrowhawk *Accipiter nisus* [10,111,112] found no clear differences between the survival rate of *Leucocytozoon* sp. among infected and uninfected birds. However, damage to kidneys during *Leucocytozoon* infection in accipitriform birds is unlikely to be inconsequential [93]. Recent observations of naturally *L. simondi*-infected anseriform birds show that they can survive infection [33,385], possibly indicating strain-related differences in virulence in different avian hosts and/or a lack of precise information about the pathology associated with early stages of the infections.

Leucocytozoid infections might not be the direct cause of disease or death, but may increase host susceptibility to predation or other disease agents, or compromise host fitness for reproduction or migration [3,386]. This issue remains insufficiently investigated, however. Little is known about the physiological and ecological costs of leucocytozoid infections, and more research on the effects of these parasites at the level of host populations is needed.

It is important to note that virus infections have been reported in some haemosporidian parasites, including *Leucocytozoon* species [4,295,387,388], but the lack of sensitive diagnostic tools has limited the understanding of these associations in wild birds. Two recent publications describe an association between haemosporidians and a new group of viruses—Matryoshka RNA viruses (MaRNAv, Narnaviridae family) [389,390]. These findings raise questions about the biological significance of such associations, but the relationships remain largely unknown. This is a major gap in knowledge given that viruses are the most abundant biological entities on Earth and certainly infect parasitic protists, including *Leucocytozoon* species [389,391]. Moreover, viruses can influence the pathogenesis of virus-infected protists, indicating the important role of parasite-virus associations in disease epidemiology. Limited available data shows that virus infections can influence virulence of some protists by increasing (hypervirulence) or decreasing (hypovirulence) disease severity in vertebrates [389,390,391]. However, nearly nothing is known about the relationships of haemosporidians with viruses. The currently available data remain mainly at the bioinformatic level. If hypovirulence occurs in haemosporidian pathogens infected with MaRNAv and the parasite becomes less virulent for vertebrates, this information could be applicable in the development of novel treatments for haemosporidiosis in the future. Widespread avian *Leucocytozoon* parasites, in which the viruses were found [2,387,389], might be used as model organisms in this innovative research.

Several diagnostic serological tests have been developed to detect antibodies against *L. caulleryi* in chickens [73,333,335,392,393,394], but they have not been developed for other *Leucocytozoon* species.

Little is known about acquired immunity during leucocytozoonosis. After recovery from primary infections of *L. caulleryi,* domestic chickens are resistant to reinfection [73]. Protection against this pathogen was achieved by immunization with recombinant vaccines, which prevent the development of megalomeronts [394]. Several different vaccines have been developed against *L. caulleryi* [333,335], but none are available for other leucocytozoids. Based on available information on *L. simondi* infections in ducks, complete immunity does not develop after sporozoite-induced infections. Reinfection is possible and can sometimes be fatal [61,357].

Treatments for leucocytozoonosis are limited, and further research and development is needed. Administration of pyrimethamine and a combination of sulfamonomethoxine and pyrimethamine is effective for treating infections with *L. caulleryi*. Clopidol is effective in reducing the numbers of gametocytes of *L. smithi* in the blood, but does not eliminate infections completely [3,72,375].

Vector control in areas where domestic turkey populations are at risk of *L. smithi* infection has been tested by applying *Bacillus thuringiensis israelensis* to streams, the source of black fly larvae. It was effective for reducing fly density, and transmission can also be used to control active transmission in areas where endangered species of birds are at risk for *Leucocytozoon* infection [4,395]. Prevention, treatment, and control of leucocytozoonosis in free-living populations of wild birds are difficult to achieve, and techniques have not been adequately developed [3,4]. Wild birds that are maintained in rehabilitation centres are at risk of infection with *Leucocytozoon*, and may experience disease and mortality if they are exposed to vectors [217]. The use of vector-proof screening is the simplest and most effective method to prevent haemosporidian infections in captive birds at sites with active transmission.

## 4. Conclusions

*Leucocytozoon* species are widespread, abundant, and diverse blood parasites, but remain the most poorly known among avian haemosporidians. They are important components of biodiversity; thus, their ecological functions and importance need further study. Important areas for research include the following. First, large genetic distances between closely related *Leucocytozoon* species and their lineages make the development of sensitive general primers difficult for the detection of parasites infecting closely related birds. This might lead to false-negatives in PCR-based diagnostics, even in blood samples that are positive by microscopy. Thus, careful selection and testing of primers for molecular diagnostics of *Leucocytozoon* species is necessary for each host group that is targeted for studies of parasite biodiversity. Microscopic examination of blood films remains essential as a control for detecting false negatives. Second, lack of diverse distinctive morphological features of blood stages (gametocytes and their host cells) prevents conclusive taxonomic work that is based solely on examination of blood films. Examination of vector and tissue stages might be helpful but remains at an infancy stage due to insufficient data. Third, co-infections of *Leucocytozoon* species and their genetic lineages are common and even predominate in avian hosts, making it difficult to accurately link DNA sequence information with the mixture of blood stages that may be observed in the same samples. As a result, molecular characterization of *Leucocytozoon* at the species level is still negligible. Fourth, the molecular characterization of described species, which have nomenclatural priority, should be an important emphasis for developing baselines for future taxonomic work. This remains a poorly explored area of research and a significant obstacle for biodiversity studies. Fifth, patterns of exo-erythrocytic development remain insufficiently understood for most described species and for the entire genus *Leucocytozoon*, but is crucial for a better understanding of the disease caused by these parasites. The application of CISH opens many new opportunities in this area of research. Sixth, difficulties in maintaining colonies of black fly vectors have hindered experimental research on vector-parasite interactions, which remains the least studied among haemosporidian parasites. Wild-caught insects should be explored more intensively in *Leucocytozoon* species vector studies. Seventh, the remarkably patchy geographical distribution of *Leucocytozoon* parasites remains unexplained among globally distributed domestic bird species despite widespread presence of susceptible vectors and suitable ecological conditions for transmission. This suggests the existence of unknown mechanisms preventing the spread of these vector-borne infections. Eighth, knowledge about the pathogenicity of these parasites remains at an infancy, preventing a better understanding of host-parasite relationships in wildlife populations. Ninth, the marked enlargement of host cell nuclei during *Leucocytozoon* infection is a unique character in haemosporidian parasites, indicating that the parasite actively transforms host cell metabolism and involves it in the production of materials needed for growth. Such host-parasite interaction is unique in haemosporidians, but remains unexplored at a molecular level. Tenth, numerous synonymous names of *Leucocytozoon* parasites are available and, where possible, should have nomenclatural priority for the description of new species. These and related questions regarding the biology of *Leucocytozoon* parasites were discussed in this review, and possible ways to address some of them were suggested.

An alarming issue that has arisen in current research on leucocytozoids and other wildlife haemosporidians is a distinct bias towards the projects that methodically collect and analyze easily detectable DNA sequence information from blood samples. Fundamental research projects on patterns of *Leucocytozoon* species development in both avian hosts and black fly vectors—particularly those using the experimental approaches—are becoming exceptionally rare. Without a larger framework based on basic life history information, an overemphasis on the collection of DNA sequence data may underestimate the true complexity of these host–parasite systems and distort our understanding of the epidemiology and pathogenicity of these parasites.

## Figures and Tables

**Figure 1 microorganisms-11-01251-f001:**
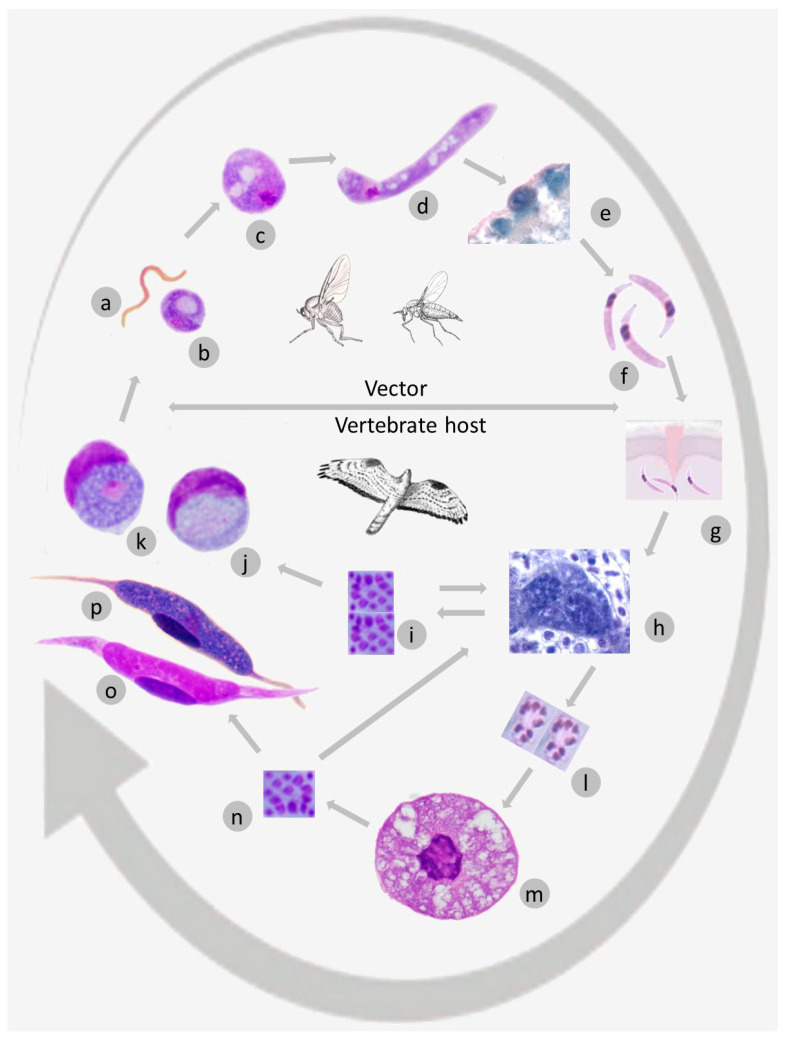
General representation of life cycle of *Leucocytozoon* parasites: microgamete (**a**), macrogamete (**b**), zygote (**c**), ookinete (**d**), oocysts (**e**), sporozoites in salivary glands of vector (**f**), sporozoite in avian host (**g**), first generation of exoerythrocytic meronts (**h**), merozoites from the first generation of meronts (**i**), microgametocyte (**j**) and macrogametocyte (**k**) in roundish host cells (**j,k**), syncytia (**l**), megalomeronts (**m**), merozoites from megalomeronts (**n**), microgametocyte (**o**) and macrogametocyte (**p**) in fusiform host cells (**o**,**p**). Double arrows (**h**,**i**) indicate possible reverse development when merozoites initiate secondary merogony in different organs.

**Figure 2 microorganisms-11-01251-f002:**
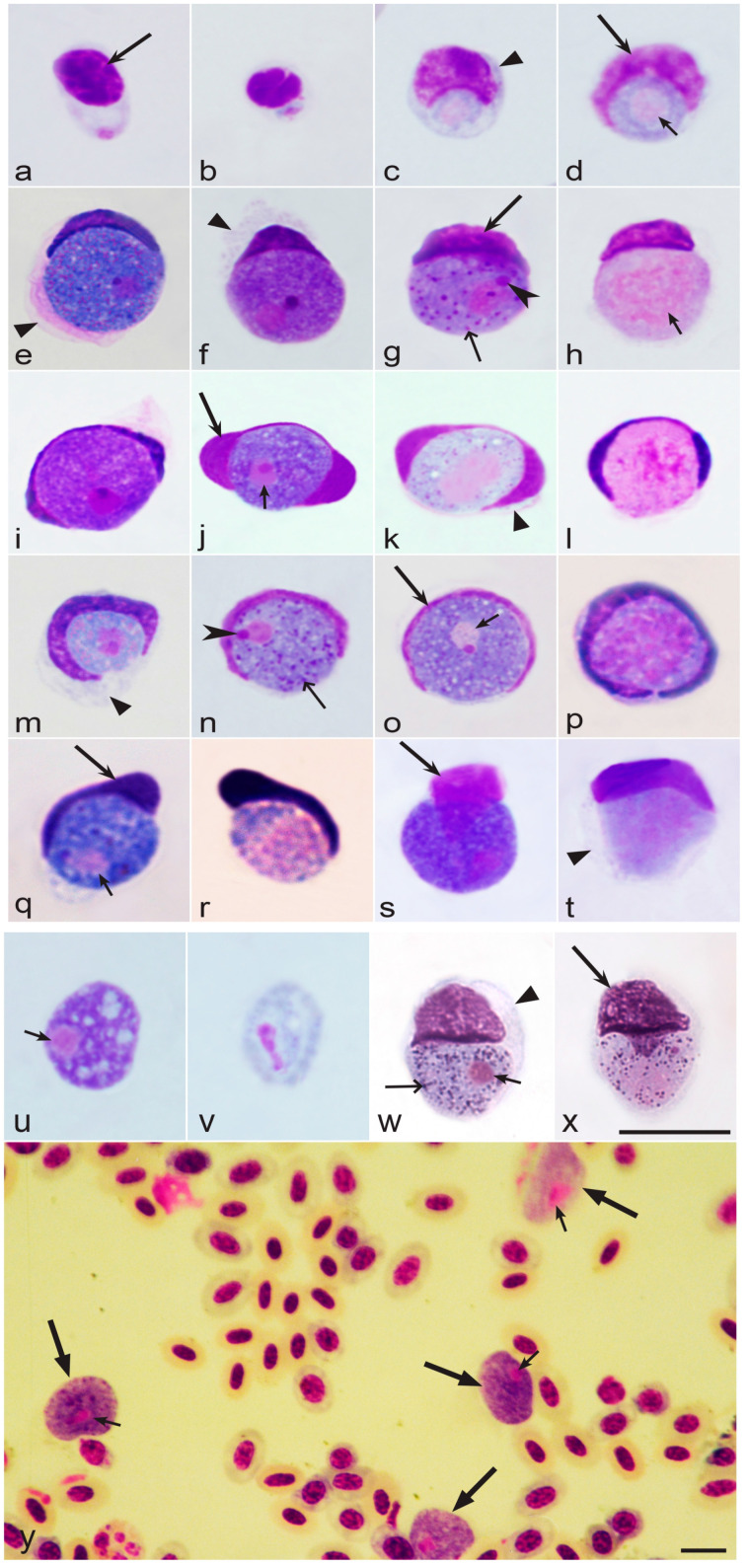
Main morphological forms (**a**–**y**) of *Leucocytozoon* (**a**–**v**,**y**) and *Saurocytozoon* (**w**,**x**) macrogametocytes (**e**–**g**,**i**,**j**,**m**–**o**,**q**,**s**,**u**,**w**,**y**) and microgametocytes (**h**,**k**,**l**,**p**,**r**,**t**,**v**,**x**,**y**), which develop in roundish host cells: merozoite in a thrombocyte (**a**); young growing gametocytes (**b**–**d**); fully grown gametocytes (**e**–**y**). Note that even youngest gametocytes markedly deform both the host cells and their nuclei (**b**–**d**), resulting in challenges to identifying their origin using morphological characters. Gametocytes are closely appressed to host cell nuclei (**b**–**t**,**w**,**x**), which assume species-specific form in different parasites. Host cell nuclei (HCN) usually extend less than 1/2 of the circumference when gametocytes of *L. fringillinarum* group (**e**–**h**) are present, but HCN cover more than ½ of the circumference of gametocytes in *L. dubreuili* (**i**–**l**) and *L. majoris* (**m**–**p**) species groups. The form of HCN is also markedly different in different species; they assume various cap-like shapes during *L. fringillinarum* infection (**e**–**h**), but are dumbbell-shaped (with more or less visible thickenings at both ends) in *L. dubreuili* (**i**–**l**), or look like narrow bands having approximately the same width along their entire lengths in *L. majoris* (**m**–**p**). HCN assume comma-like shapes during *L. quynzae* infection (**q**,**r**), but are markedly variable in form and often assume positions above the gametocytes during *L. californicus* infection (**s**,**t**). Fully grown gametocytes of *Leucocytozoon bennetti* (**u**,**v**) and *L. (Akiba) caulleryi* (**y**) often enucleate host cells and look like naked parasites in blood films. Intense parasitemia of L. *(A.) caulleryi* is shown (**y**), with three macrogametocytes (bottom) and one microgametocyte (top) present. Morphology of *S. tupinambi* (**w**,**x**) gametocytes and its host cell are similar to *Leucocytozoon* species (see (**e**–**h**)). The cytoplasm of host cells is usually present around growing gametocytes as more or less evident and pale margins of variable form (**c**,**e**,**f**,**k**,**m**,**t**,**w**), but is usually invisible in host cells containing fully grown gametocytes (**g**,**p**,**r**,**s**,**u**,**v**). Sexual dimorphic characters—the densely stained cytoplasm and the small compact nuclei with visible nucleolus—are features of macrogametocytes, but not of microgametocytes (compare figures (**j,k**)). Long simple arrows—host cell nuclei. Triangle arrowheads—remnants of host cell cytoplasm. Short simple arrows—parasite nuclei. Simple arrowheads—nucleoli. Simple wide long arrows—volutin. Triangle long arrows—gametocytes. Other explanations are given in the text. Bars = 10 µm.

**Figure 3 microorganisms-11-01251-f003:**
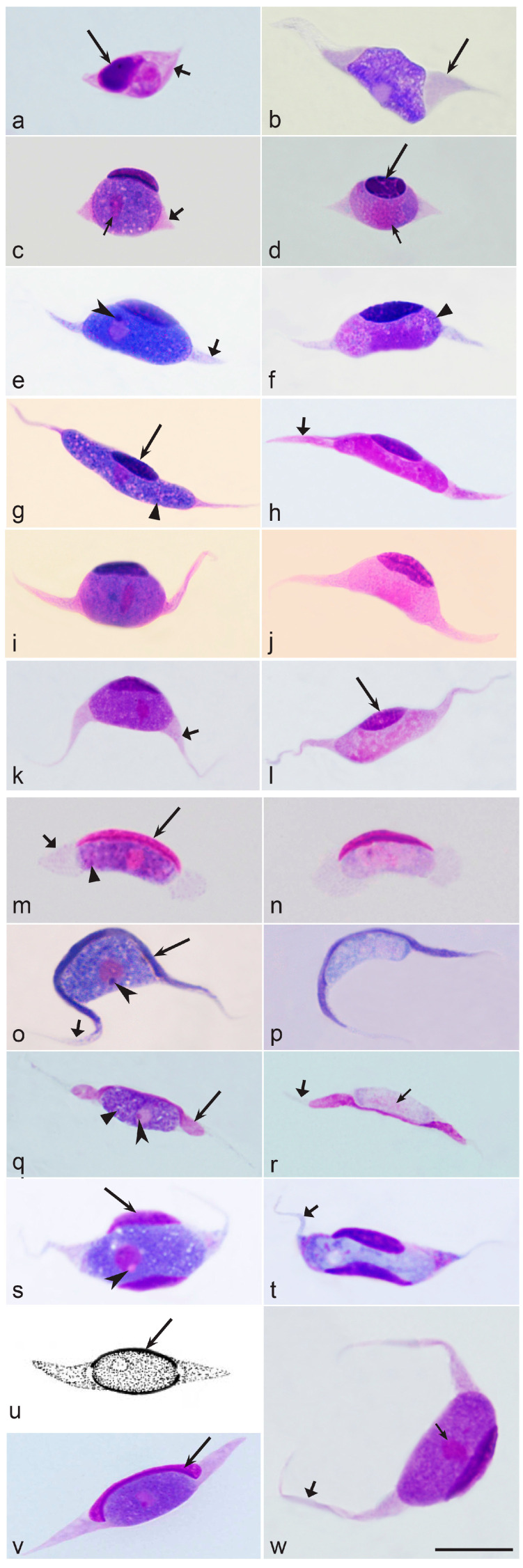
Main morphological forms (**a**–**w**) of *Leucocytozoon* macrogametocytes (**b**,**c**,**e**,**g**,**i**,**k**,**m**,**o**,**q**,**s**,**u**,**v**,**w**) and microgametocytes (**d**,**f**,**h**,**j**,**l**,**n**,**p**,**r**,**t**), which develop in fusiform host cells: young growing gametocyte (**a**) and fully grown gametocytes (**b**–**w**) of *L. hamiltoni* (**b**), *L. mathisi* (**c**–**f**), *L. buteonis* (**g**–**l**), *L. grallariae* (**m**,**n**), *L. pterotenuis* (**o**,**p**), *L. simondi* (**q**,**r**), *L. smithi* (**s**,**t**), *L. maccluri* (**u**), *L. danilewskyi* (**v**), and *L. lovati* (**w**). Note that very young gametocytes (**a**) induce the elongation of host cells and development of cytoplasmic fusiform processors (CFP). Form of gametocytes is relatively similar in same and different parasite species—it varies from more or less roundish to oval—but the parasites can be readily distinguished due to length and shape of CFP, and especially due to the different shape of the host cell nucleus (HCN) and its position in regard to CFP. For example, in *L. mathisi, L. buteonis*—the parasites of *L. toddi* group—the HCN assumes a cap-like or almond-like shape or resembles the nucleus of uninfected erythrocyte, and it extends less than 1/3 of the circumference of the gametocyte (**c**–**l**). Based on these features, *L. mathisi* (**c**–**f**) and *L. buteonis* (**g**–**l**) are hardly distinguishable, but the form of host cell CFP is different in these two species—the CFP processors are longer in *L. buteonis* (**g**–**l**) than in *L. mathisi* (**c**–**f**). *L. lovati* induces development, especially thin and long CFP (**w**). HCN assume a unique form in *L. hamiltoni* ((**b**) the HCN is split into two more or less symmetrical portions, each located at an end of the host cell within its elongate CFP), *L. grallariae* ((**m**,**n**) the HCN assume slender waning moon-like form, and can reach CFP, but never extended into CFP), *L. pterotenuis* ((**o**,**p**) the HCN is band-like and markedly extended into CFP), *L. simondi* ((**q**,**r**) the HCN assume a band-like shape with clear dumbbell-like thickenings on both ends which extend into CFP and do not adhere to gametocyte), *L. smithi* ((**s**,**t**) the HCN is split into two portions which locate on each side of gametocyte opposite to each other and never extend into CFP), *L. maccluri* ((**u**) the HCN is uniformly dispersed as a narrow band around nearly all of the circumference of gametocyte), *L. danilewskyi* ((**v**) the HCN is more or less dumbbell-shaped with clear thickenings at both ends, which are appressed to gametocyte, but do not extend into CFP). Sexual dimorphic characters—the densely stained cytoplasm and the small compact nucleus with visible nucleolus—are features of macrogametocytes, but not of microgametocytes (compare figures (**e**,**f**)). Long simple arrows—host cell nuclei. Triangle wide short arrows—fusiform processors of host cell. Short simple arrows—parasite nuclei. Simple arrowheads—nucleoli. Triangle arrowheads—volutin. Other explanations are given in the text. Bar = 10 µm.

**Figure 4 microorganisms-11-01251-f004:**
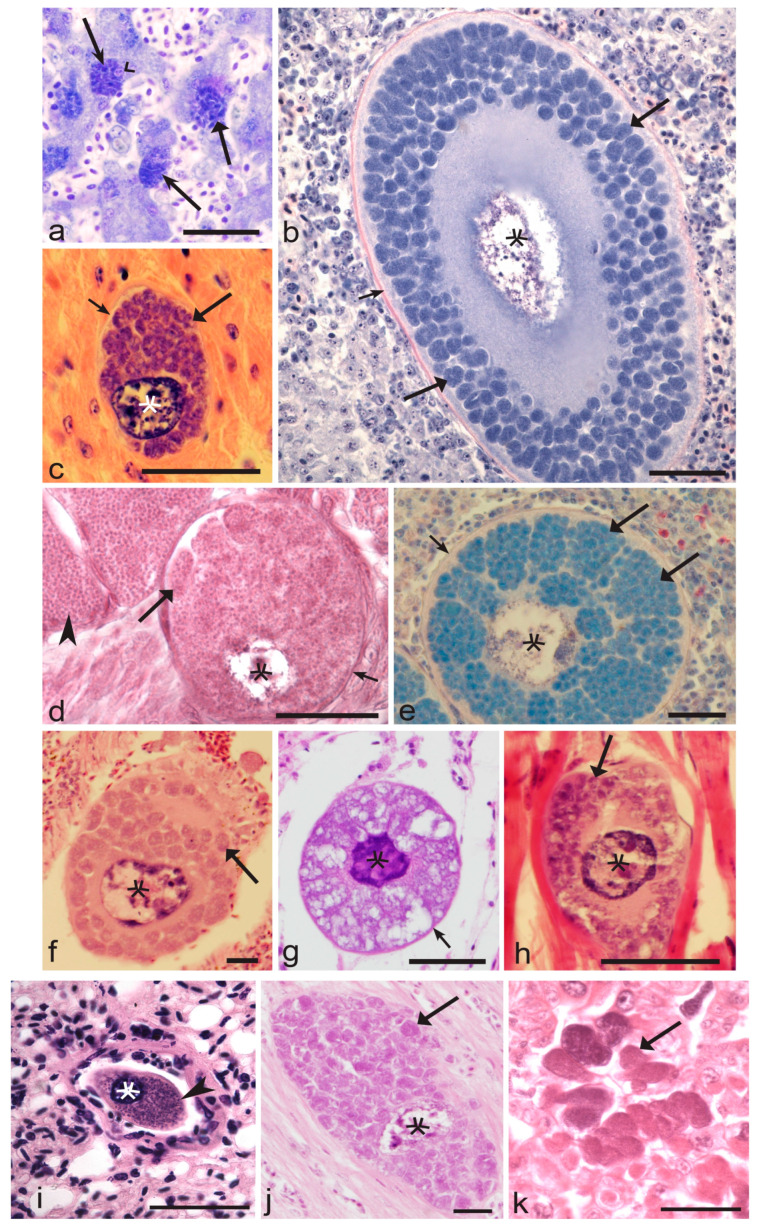
Exo-erythrocytic stages of *Leucocytozoon* parasites: meronts (**a**) and megalomeronts (**b**–**k**). Hepatic meronts of *L. simondi* in a liver section (**a**). Note the numerous meronts and the much haemorrhagic and disorganized liver tissue; nuclei visible and non-changed in size, in comparison to nuclei size in non-infected hepatocytes. Megalomeronts of *L. sakharoffi* (**b**,**e**), *L. simondi* (**c**,**d**,**i**,**j**), *L.* sp. (**f**–**h**), and *L. (Akiba) caulleryi* (**k**) in different stages of maturation in liver (**a**,**b**), heart (**c**,**j**), spleen (**e**,**f**), lungs (**g**,**i**), kidney (**k**), muscle (**d**,**h**). Note markedly enlarged host cell nuclei or the ‘central bodies’ (**b**–**j**), more or less evident capsular-like wall surrounding the megalomeronts (**b**–**j**), numerous variously shaped cytomeres (**b**–**j**), and mature merozoites (**d**,**i**). Simple long arrows—hepatic meronts. Simple wide arrowhead—nucleus of infected hepatocyte. Triangle wide long arrows—cytomeres. Simple short arrows—wall of megalomeronts. Simple arrowheads—merozoites. Stars—host cell nuclei of megalomeronts. Bars = 50 µm.

**Table 1 microorganisms-11-01251-t001:** Mitochondrial DNA sequences, which have been developed for molecular detection and identification (barcoding) of avian *Leucocytozoon* parasites.

Parasite Species ^1^	GenBank Accession and Lineage Code (in Parentheses) ^2^	Reference ^3^
*L. balmorali* (R, F)	Not available	[4,122]
*L. bennetti* (R)	Not available	[4,123]
*L. berestneffi* (R)	Not available	[4,64,85,90]
*L. buteonis* (F) ^4^	DQ177253 (BUBT2), DQ177273 (BUTJAM10), DQ177264 (BUTREG01)	[4,124,125]
*L. californicus* (R)	FASPA02 (KR422359)	[20]
*L. caprimulgi* (R, F)	Not available	[4,126]
*L. caulleryi* (R)	GALLUS05 (AB302215), complete mitochondrial DNA genome	[4,127]
*L. centropi* (R)	Not available	[4,128,129]
*L. cheissini* (F)	Not available	[4,48]
*L. colius* (R)	Not available	[4,130]
*L. communis* (R)	Not available	[4,131]
*L. danilewskyi* (R, F) ^5^	Not available	[4,124,132]
*L. dizini* (R)	Not available	[4,133]
*L. dubreuili* (R) ^6^	TUMER09 (KY653795)	[4,90,134,135]
*L. eurystomi* (R, R)	Not available	[4,126]
*L. fringillinarum* (R) ^7^	Not available	[4,136]
*L. grallariae* (F)	GRSQU01 (MK103895), partial mitochondrial DNA genome	[9]
*L. grusi* (R, F)	Not available	[4,137]
*L. hamiltoni* (F)	Not available	[4,138]
*L. leboeufi* (R)	Not available	[4,139]
*L. legeri* (R)	Not available	[4,140]
*L. lovati* (R, F) ^8^	LAMUT01 (AB183550)	[4,56,141,142]
*L. maccluri* (R, F)	Not available	[4,143]
*L. macleani* (R, F) ^9^	Not available	[4,30,64,144]
*L. majoris* (R) ^10^	Not available	[4,145]
*L. marchouxi* (R)	Not available	[4,146,147]
*L. mathisi* (F) ^4^	ACNI04 (DQ177252), ACCOP01 (DQ177250)	[125,140]
*L. neavei* (F)	Not available	[4,126,148]
*L. neotropicalis* (F)	PIRIE01 (MK103894), partial mitochondrial DNA genome	[9]
*L. nycticoraxi* (R)	Not available	[4,149]
*L. nyctyornis* (R)	Not available	[4,150]
*L. podargii* (R)	Isolate WA (MK358451)	[151,152]
*L. polynuclearis* (R)	COLAUR01 (MW626894), DRYALB01 (MW626892)	[21]
*L. pterotenuis* (F) ^11^	(KM610046) partial mitochondrial DNA genome, GRARUF01 (KM272250)	[9,153]
*L. quynzae* (R)	HEAME01 (KF479480) partial mitochondrial DNA genome, HELIAM01 (KF309188)	[18]
*L. sakharoffi* (R)	Not available	[4,64,90]
*L. schoutedeni* (R)	GALLUS06 (DQ676823), GALLUS07 (DQ676824)	[4,29,154]
*L. simondi* (R, F) ^12^	Not available	[4,146,155,156]
*L. smithi* (R, F)	Not available	[4,157]
*L. sousadiasi* (F)	Not available	[4,133]
*L. squamatus* (R)	Not available	[4,158]
*L. struthionis* (R)	Not available	[4,159]
*L. tawaki* (R)	Not available	[4,160]
*L. toddi* (F, occationally R) ^13^	Not available	[4,64,161]
*L. vandenbrandeni* (R)	Not available	[4,162]

^1^ Mainly, the readily distinguishable morphospecies were included in this table. Letters (in parentheses) indicate types of host cells—roundish (R) and fusiform (F)—in which gametocytes of the corresponding species were found. Synonymous names of the morphologically indistinguishable parasites were grounded and summarised in [4]; these data are extensive and not repeated in this review. The following additional synonymous names should be considered: *L. artamidis* (possible synonym of *L. sakharoffi* group [163]), *L. atkinsoni* and *L. greineri* (possible synonyms of *L. fringillinarum* group, [164,165]), *L. coracinae* (possible synonym of *L. majoris* group [166], *L. pogoniuli* and *L. trachyphoni* (possible synonyms of *L. squamatus* group) [167]). ^2^ Mainly DNA sequences, for which the parasite species identity was supported by morphological analysis were included in this table. In cases of the identical DNA sequences of the same morphospecies, the preference was given to the GenBank accessions, which directed a reader straight to articles containing morphological parasite descriptions. Using such GenBank information simplifies the parasite identification. Where possible, the codes of lineages were given according to the MalAvi database. ^3^ References of the articles containing a description of parasite morphology and/or discussion on their molecular characterization, which are valuable for species identification. The original parasite descriptions and reviews were cited preferably. ^4^ Both *L. buteonis* and *L. mathisi* belong to the *L. toddi* group [93]. The first two species are readily distinguishable from each other due to the different length and form of the host cell cytoplasmic processors, which are significantly shorter in *L. mathisi* (compare Figure 3c–f with Figure 3g–l) [125]. ^5^ Numerous *Leucocytozoon* lineages were found in owls (Strigiformes) worldwide [19], however, the molecular characterization of *L. danilewskyi* (synonym is *L. ziemanni*, see [168]) from its type vertebrate host *Athene noctuae* is absent. The lineage BUBO01 (EU624137) found in *Bubo bubo* belongs to the *L. danilewskyi* group [169], which also includes the similar lineages recovered from other Strigiformes species—ASOT1 (EF607286), STAL1 (EF607285), ASOT2 (EF607284), and many others. However, this group of lineages might also include several cryptic species, which can be named in the future. Thus, the detailed unravelling of the parasites belonging to the *L. danilewskyi* group and the determination of the relationships between them require the initial molecular characterization of the parasite inhabiting its type hosts sampled close to the type locality—*A. noctuae* in Europe. Blood stages of the lineage ASOT3 (KY653781), which was found in *Asio otus* (Strigidae) in Europe, are similar to the blood stages seen in *A. noctuae.* Moreover, the area of transmission of leucocytozoids in these two species of owls often overlaps, so the lineage ASOT3 is probably closely related to the lineage of *L. danilewskyi* in *A. noctuae.*
^6^ Gametocytes of the lineage TUMER09 (KY653795) were found in *Turdus merula* in Europe [8]; they showed all main diagnostic characters of *L. dubreuili* from the type host *Turdus iliacus* [G. Valkiūnas, per. obs.]. However, the molecular characterization of *L. dubreuili* from its probable type vertebrate host *Turdus iliacus* (Turdidae) at the type locality (Eastern Asia) remains preferable for the certain unravelling of the parasites belonging to the *L. dubreuili* group and the determination of the relationships between them. The group of *L. dubreuili* morphospecies—even the parasites of the closely related Turdidae birds—might include several cryptic species, which can be named in the future. ^7^ Molecular characterization of *L. fringillinarum* from its type vertebrate host *Fringilla coelebs* (Fringillidae) and the type locality (Europe) is absent. The lineage CB1 (FJ168564) was attributed to *L. fringillinarum* [170], but it was detected from the non-type host (*Pipilo chlorurus*, Passerellidae) far away from the type locality (North America). The lineages TFUS01 (JQ815432), TFUS02 (JQ815433), TFUS03 (JQ815434), and TFUS04 (JQ815435) were also attributed to *L. fringillinarum* [171], but they were also detected from the non-type host (*Turdus fuscater*, Turdidae) far away from the type locality (South America). Blood stages of all these parasites are similar to *L. fringillinarum*; however, it remains unclear if these lineages can be genetically attributed to *L. fringillinarum*. Detailed unravelling of the parasites belonging to *L. fringillinarum* group requires the initial molecular characterization of the parasite inhabiting *F. coelebs* in Europe. Blood stages of the lineage SISKIN2 (AY393796), which was found in *Carduelis spinus* (Fringillidae) in Europe, are indistinguishable from blood stages seen in *F. coelebs*; and the areas of transmission of *L. fringillinarum*-like parasites often overlap in these birds, so the lineage SISKIN2 (AY393796) is probably closely related to the type lineages of *L. fringillinarum* from *F. coelebs*. ^8^ The morphology of blood stages seen in *Lagopus muta* (LAMUT01, AB183550) [56,142] is indistinguishable from *L. lovati* in its type host *Lagopus scoticus*, so this lineage probably belongs to this species. However, the molecular characterization of the same morphospecies from its type host *L. scoticus* in Western Europe (the type locality) is still needed for the determination of a barcoding DNA sequence for *L. lovati*, and further specification of the parasites of the *L. lovati* group. ^9^ Molecular characterization of *Leucocytozoon macleani*—the parasite of *Phasianus colchicus* (Phasianidae)—is absent. However, numerous lineages of the morphologically similar parasite developing in the fusiform host cells were found in domestic chicken (*Gallus gallus*, Phasianidae) [30] and attributed to *Leucocytozoon sabrazesi* (possible synonym of *L. macleani*). The barcoding sequence GALLUS08 (AB299369) of *L. sabrazesi* might belong to the *Leucocytozoon macleani* group. However, the certain unravelling of the parasites belonging to this group primarily requires developing molecular characterization of the parasite inhabiting *Ph. colchicus*. ^10^ Molecular characterization of *L. majoris* from its type vertebrate host *Parus major* (Paridae) is absent. The lineage ZOLEU02 (FJ168563) was attributed to *L. majoris* [170], but it was detected from the non-type host (*Zonotrichia leucophrys*, Passerellidae) far away from the type locality (North America). The detailed unravelling of the parasites belonging to the *L. majoris* group primarily requires the molecular characterization of the parasite inhabiting *P. major* in Europe. ^11^ Description of *L. pterotenuis* is valid in part—for the gametocytes developing in fusiform host cells, but not for the gametocytes developing in roundish host cells [9]. ^12^ Molecular characterization of *Leucocytozoon simondi*—the parasite of Anseriformes birds—is absent from its type host *Anas crecca*. This bird is widespread in Eurasia, with six *Leucocytozoon* lineages reported in this avian host [19]. Numerous lineages of *Leucocytozoon* parasites were found in ducks [34,41,172], and they might belong to the *L. simondi* group. Detailed unravelling of the taxonomic position of the parasites belonging to this group primarily requires the molecular characterization of the parasite inhabiting *A. crecca*. The group of *L. simondi* morphospecies—even the parasites of the closely related Anatidae birds—might include cryptic species, which can be named in the future. ^13^ *Leucocytozoon toddi* was originally described from *Kaupifalco monogrammicus* (Accipitridae) in Central Africa (the Democratic Republic of the Congo). The lineage ACCFRA01 (AY762076) was attributed to *L. toddi* [173]. The well-illustrated blood stages [173] showed that this parasite certainly belongs to the *L. toddi* group, but the DNA sequence came from the non-type host *Accipiter francesii* (Accipitridae) sampled in Madagascar. The detailed unravelling of the genetically markedly diverse parasites of the *L. toddi* group [93] primarily requires the molecular characterization of the parasite inhabiting *K. monogrammicus* in Africa. The group of *L. toddi* morphospecies—even the parasites of the closely related Accipitridae birds—certainly includes several cryptic species [174], which can be named in the future. Gametocytes in the roundish host cells were occasionally seen together with gametocytes in the fusiform host cells in the same blood films of accipitriform birds [4]. Due to the exceptionally rare such records in species of Accipitriformes, the gametocytes in the roundish host cells might belong to other non-described species occurring in co-infection.

## Data Availability

Main data were presented in this article and can be provided on request.

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
