# Peer review of "Insights into the Biology of Leucocytozoon Species (Haemosporida, Leucocytozoidae): Why Is There Slow Research Progress on Agents of Leucocytozoonosis?"

_microorganisms, 2023, doi:10.3390/microorganisms11051251_

Round 1

Reviewer 1 Report

In the review manuscript entitled “Insights into the biology of Leucocytozoon species (Haemosporida, Leucocytozoidae): Why is there slow research progress on agents of Leucocytozoonosis?” the authors discuss the plausible reasons for slow pace of research progress on Leucocytozoon species and possible solutions to address the limitations. Below are the suggestions to improve the manuscript.

1.    Lines #102-103: Drawings were also provided to illustrate morphological features of the parasites for…..

The spelling errors should be identified and corrected throughout the manuscript.

2.    The authors should include appropriate highlights/tables/bullet point summaries in the manuscript to make it easier for the readers.

The quality of English is good. There are some spelling errors and they should be corrected.

Author Response

Reviewer 1

In the review manuscript entitled “Insights into the biology of Leucocytozoon species (Haemosporida, Leucocytozoidae): Why is there slow research progress on agents of Leucocytozoonosis?” the authors discuss the plausible reasons for slow pace of research progress on Leucocytozoon species and possible solutions to address the limitations. Below are the suggestions to improve the manuscript.

  1. Lines #102-103: Drawings were also provided to illustrate morphological features of the parasites for…..

The spelling errors should be identified and corrected throughout the manuscript.

Corrected. We also read the text carefully again and a few spelling arrows were recognised and corrected. This text was also read by dr. Carter Atkinson (USA) who kindly gave as a hand with English language editing before initial submission.

  1. The authors should include appropriate highlights/tables/bullet point summaries in the manuscript to make it easier for the readers.

Sorry, this suggestion was not very clear to us. The text is divided into numerous sections/subsections, which titles bullet the main idea of each section. Each section/subsection contains conclusive paragraphs or statements summarising the main/important result, which worth of attention. The conclusion section again summarises the main ideas of the review point by point; these short summaries direct reader to the corresponding review chapters and should be helpful for readers. If possible, we should maintain the main text as it is.

Reviewer 2 Report

The submitted manuscript presents a review whose main aim is to summarize available knowledge and recent findings on the biology of avian Leucocytozoon species.

The review presents an updated state-of-the-art on the subject with manuscript sections including an introductory section in which it is mentioned that while much knowledge exists and several reviews have been published on various aspects of Leucocytozoon species biology, these have not been updated in the past 15 years and thus, this review discusses available basic information on the Leukocytozoon species, and a timely and updated review on the increasing expansion of Leucocytozoon infection in birds, mainly associated with climate change.

Major headings of the manuscript describe the Leucocytozoon life cycle with remarks on some unique haemosporidian life cycle features; the diversity of Leucocytozoon species with discussion on the main classification problems, the Leucocytozoon species taxonomy, the problem of synonymous species names, and the morphological characters of blood stages. Additional headings include Host cells of Leucocytozoon parasites; The obstacles in research on exo-erythrocytic development; Challenges of the molecular characterization of Leucocytozoon parasites; The obstacles in Leucocytozoon parasite vector Research; Puzzles of the geographical distribution; Pathogenicity and related health and the host-parasite association issues.

Overall, the review is well written, summarizes the literature published on the topic, discusses it critically, identifies major methodological problems, and points out existing research gaps.

Minor corrections that authors need to take care of are as follows

Line 63. Add a period before ‘Thus’

Line 102. Correct ‘Grawings´

Line 229. Add ‘are’ before ‘present’

Line 239. Correct punctuation marks in: ,,naked”

Line 286. Should it be ‘extended’ in ‘markedly extends into CFP’

Line 156. Could use ‘lymph nodes’ instead of ‘lymph glands’

Line 163. Could use ‘dormant’, instead of "sleeping" tissue stages?

Line 170. Could use ‘capsule-like’ instead of ‘capsular-like’

Line 204. Correct the sentence ‘Gametocytes possess sexual potency as they produce gametes’, as sexual potency can be defined as the ability to carry out and consummate sexual intercourse.

Line 375. Correct ‘names’ in ‘which can be names in future’.

Line 440. Correct ‘names’ in ‘which can be names in future’.

Line 449. Spell DNR, first time used

Line 577. Reference 215, should it be 216? Otherwise, reference 216 is not cited in the text

Line 652. Add ‘of’ after ‘use’ in ‘use the available synonymous’

Line 1103. Add ‘is’ in between ‘It worth noting’

Line 1341. Add ‘of’ in-between ‘characterization Leucocytozoon’  

Line 1356. Correct the term ‘vector-born’

Line 1993. Reference 289 does not appear cited in the text. Please check

Line 2066. Reference 323 does not appear cited in the text. Please check 

Author Response

Reviewer 2

The submitted manuscript presents a review whose main aim is to summarize available knowledge and recent findings on the biology of avian Leucocytozoon species.

The review presents an updated state-of-the-art on the subject with manuscript sections including an introductory section in which it is mentioned that while much knowledge exists and several reviews have been published on various aspects of Leucocytozoon species biology, these have not been updated in the past 15 years and thus, this review discusses available basic information on the Leucocytozoon species, and a timely and updated review on the increasing expansion of Leucocytozoon infection in birds, mainly associated with climate change.

Major headings of the manuscript describe the Leucocytozoon life cycle with remarks on some unique haemosporidian life cycle features; the diversity of Leucocytozoon species with discussion on the main classification problems, the Leucocytozoon species taxonomy, the problem of synonymous species names, and the morphological characters of blood stages. Additional headings include Host cells of Leucocytozoon parasites; The obstacles in research on exo-erythrocytic development; Challenges of the molecular characterization of Leucocytozoon parasites; The obstacles in Leucocytozoon parasite vector Research; Puzzles of the geographical distribution; Pathogenicity and related health and the host-parasite association issues.

Comments on the Quality of English Language

Overall, the review is well written, summarizes the literature published on the topic, discusses it critically, identifies major methodological problems, and points out existing research gaps.

Thank you for high evaluation of our study

Minor corrections that authors need to take care of are as follows

Line 63. Add a period before ‘Thus’

Corrected.

Line 102. Correct ‘Grawings´

Corrected

Line 229. Add ‘are’ before ‘present’

Corrected.

Line 239. Correct punctuation marks in: ,,naked”

Corrected.

Line 286. Should it be ‘extended’ in ‘markedly extends into CFP’

Corrected.

Line 156. Could use ‘lymph nodes’ instead of ‘lymph glands’

Corrected. Many thanks for this note!

Line 163. Could use ‘dormant’, instead of "sleeping" tissue stages?

Corrected.

Line 170. Could use ‘capsule-like’ instead of ‘capsular-like’

Corrected.

Line 204. Correct the sentence ‘Gametocytes possess sexual potency as they produce gametes’, as sexual potency can be defined as the ability to carry out and consummate sexual intercourse.

Corrected as ‘Gametocytes are responsible for production of gametes...‘

Line 375. Correct ‘names’ in ‘which can be names in future’.

Corrected as ‘which can be named…’.

Line 440. Correct ‘names’ in ‘which can be names in future’.

Corrected as ‘which can be named’.

Line 449. Spell DNR, first time used

It should be DNA. Corrected and spelled (l. 106).

Line 577. Reference 215, should it be 216? Otherwise, reference 216 is not cited in the text

Thank you for this note. The reference 216 is now added on line 594.

Line 652. Add ‘of’ after ‘use’ in ‘use the available synonymous’

Corrected.

Line 1103. Add ‘is’ in between ‘It worth noting’

Corrected.

Line 1341. Add ‘of’ in-between ‘characterization Leucocytozoon’  

Corrected.

Line 1356. Correct the term ‘vector-born’

Corrected.

Line 1993. Reference 289 does not appear cited in the text. Please check

Thank you for this comment. This reference was added (line 1026).

Line 2066. Reference 323 does not appear cited in the text. Please check 

Thank you for this comment. This reference was added (line 1091).